# Unleashing the Power of Deep Dehazing Models: A Physics-guided Parametric Augmentation Net for Image Rehazing

## Abstract

Image dehazing faces significant challenges in real-world scenarios due to the large domain gap between synthetic and real-world hazy images, which often hinders dehazing performance. Collecting real-world datasets is particularly difficult, as hazy and clean image pairs must be captured under identical conditions. To address this, we propose a Physics-guided Parametric Augmentation Network (PANet) that generates realistic hazy and clean training pairs, enhancing dehazing performance in real-world applications. PANet consists of two components: a Haze-to-Parameter Mapper (HPM), which projects hazy images into a parametric space representing haze characteristics, and a Parameter-to-Haze Mapper (PHM), which converts resampled haze parameters back into hazy images. By resampling individual haze parameter maps at the pixel level in the parametric space, PANet generates diverse hazy images with physically explainable haze conditions that are not present in the training data. Our experimental results show that PANet effectively enriches existing hazy image benchmarks, significantly improving the performance of current dehazing models.

## 1 Introduction

Images captured in hazy environments often experience significant degradation, resulting in poor contrast and distorted appearances. In real-world scenarios, these hazy artifacts tend to be dense and non-uniform, severely affecting both the visual quality and the visibility of scenes. This degradation also negatively impacts downstream computer vision tasks such as object detection, tracking, and scene understanding. Image dehazing seeks to recover high-quality, clear images from single hazy inputs. However, this task is a highly ill-posed inverse problem, made challenging by the substantial information loss caused by haze-induced degradation.

Recently, image dehazing techniques have seen significant advancements, largely driven by the success of deep learning. Numerous studies (Liu et al., 2019; Deng et al., 2020; Cui et al., 2023; Guo et al., 2022; Song et al., 2023; Yu et al., 2022; Li et al., 2019b; Qu et al., 2019; Wu et al., 2021) have focused on enhancing dehazing performance through innovative network architecture designs. Many of these works leverage CNN-based modules to learn haze-specific features, employing techniques such as channel-wise attention (Liu et al., 2019), haze-aware feature distillation (Deng et al., 2020), and dual-domain selection (Cui et al., 2023). Additionally, inspired by the success of Transformers (Vaswani et al., 2017) in various vision tasks (Dosovitskiy et al., 2021; Chen et al., 2021a; Ranftl et al., 2021), several recent studies have adopted Transformer-based architectures for image dehazing. Examples include transmission-aware Transformers (Guo et al., 2022) and window-based Transformers (Song et al., 2023), further pushing the boundaries of dehazing performance with their enhanced feature extraction and attention mechanisms.

These methods predominantly rely on synthetic hazy image datasets (Li et al., 2019a), which are generated using physical scattering models (McCartney, 1976; Nayar & Narasimhan, 1999; Narasimhan & Nayar, 2003) to produce homogeneous synthetic hazy images:

$$I(z) = J(z)t(z) + A(1 - t(z)),$$
$$t(z) = e^{-\beta d(z)}, \tag{1}$$

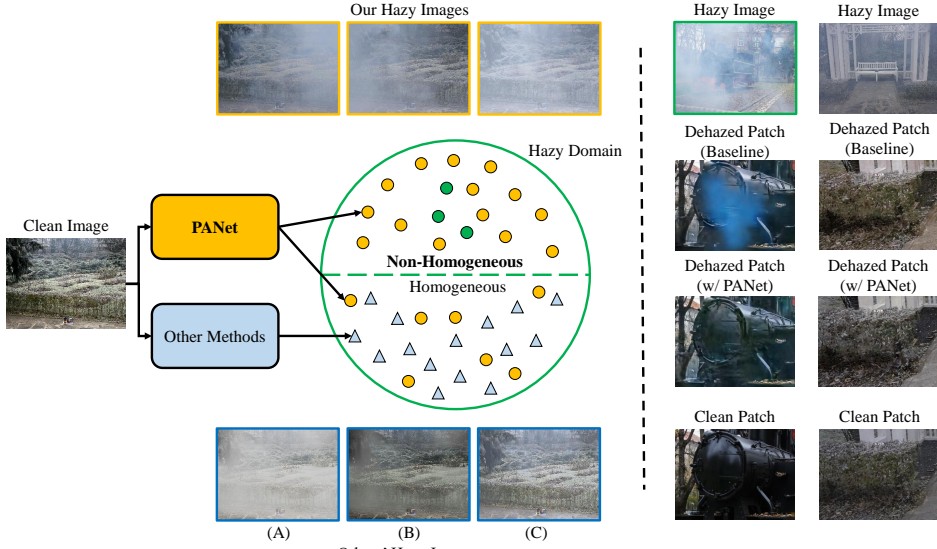

Figure 1: Left: Examples of augmented images produced by PANet (in yellow rectangles) compared to other augmentation methods (in blue rectangles). Existing non-homogeneous hazy datasets Ancuti et al. (2020; 2021) offer only a limited number of training pairs, as indicated by the green circles. Previous augmentation techniques Wu et al. (2023); Yang et al. (2022) struggle to generate effective non-homogeneous hazy images. In contrast, PANet is capable of producing both realistic non-homogeneous and homogeneous hazy images, as demonstrated in the yellow rectangles. Right: A comparison of dehazing results with and without using data augmented by PANet.

where $I(z)$ and $J(z)$ denote the hazy image and its clean version, $A$ and $t(z)$ are the atmospheric light and transmission map, $\beta$ and $d(x)$ are the haze density and depth map, and $z$ is the pixel index.

While physical scattering models can generate abundant pairs of hazy and clean images, the significant domain gap between synthetic and real-world hazy image distributions often hampers dehazing performance in practical settings (Gui et al., 2023; Zhang et al., 2021; Chen et al., 2021b). Real-world hazy images typically exhibit dense and non-homogeneous haze (Ancuti et al., 2019; 2020; 2021), which synthetic models struggle to replicate, as shown in Figure 1. This discrepancy limits the effectiveness of dehazing models trained on synthetic data when applied to real-world conditions.

To tackle this issue, existing works (Ancuti et al., 2020; 2021) have focused on collecting real-world non-homogeneous hazy and clean image pairs for training. However, gathering such datasets is both difficult and expensive, as it requires capturing both hazy and clean images under identical conditions, including matching moving objects and consistent background lighting. As a result, these datasets are typically limited in size, which significantly constrains the performance of deep dehazing models in real-world applications. Some methods have attempted to enhance the diversity of hazy images via brightness adjustments (Wu et al., 2023) or global haze density adjustments (Yang et al., 2022), as illustrated in Figs. 1(B) and 1(C). Nevertheless, they ignore the above important fact that real-world haze distributions are often dense and non-homogeneous, making the domain gap still large. Therefore, it is crucial to propose a new approach to learn to generate additional realistic non-homogeneous hazy images with various haze conditions from existing hazy and clean image pairs without heavily relying on a high-cost data collection process.

In this paper, we propose a novel Physics-guided Parametric Augmentation Network (PANet) designed to effectively augment realistic hazy and clean image pairs, thereby improving dehazing performance in real-world scenarios. PANet employs a Haze-to-Parameter Mapper (HPM) to project the hazy image from a clean and hazy training pair into a parametric space defined by haze characteristics. This is followed by a Parameter-to-Haze Mapper (PHM), which augments the haze parameters within the parametric space and then uses these augmented parameters to generate new hazy images, enriching the dataset with realistic variations of haze. Specifically, inspired by the widely-adopted Physical Scattering Model (PSM) in Eq. (1), given a hazy/clean image pair, HPM parameterizes the hazy image into two pixel-wise maps: haze density and atmospheric light. These

estimated parameters are then modified (or fixed) and used to translate the clean image into its hazy versions with various haze patterns in a two-step process. First, we apply the haze parameters to the PSM model in Eq. (1) to create an initial, reasonable hazy image. In the second step, we employ a Data-driven Haze Refiner (DHR) $N_{\mathrm{DHR}}(\cdot)$ to refine this initial image, enhancing its realism. Despite the guidance from physics, retrieving accurate pixel-wise haze parameters from a clean and hazy image pair remains a highly ill-posed problem, especially when dealing with dense or non-homogeneous haze, which can cause severe occlusion and distortion. This inherent ill-posedness often results in inaccurate parameter estimation, leading to suboptimal visual quality when relying solely on PSM-based generation (Gui et al., 2023; Zhang et al., 2021; Chen et al., 2021b). To mitigate this issue, we integrate DHR to refine the hazy image and employ the reconstruction error as supervision, forming a cyclic haze-parameter-haze learning process that combines both HPM and DHR, as detailed in Sec. 3.

PANet offers several unique advantages. First, by utilizing scattering model-based physics guidance, the estimated haze parameters retain physical significance, making both the parametric augmentation process and the resulting hazy image generation explainable. Second, our haze-parameter-haze mapping framework establishes a cyclic learning process for haze generation, involving haze parameter estimation, parametric augmentation, model-based initialization, and data-driven refinement. This cyclic learning leads to more accurate haze parameter estimation and realistic hazy image generation, addressing inaccuracies in model-based methods while simplifying the design of purely data-driven deep models through meaningful initialization based on scattering models. Third, by resampling these explainable haze parameters, we can easily generate augmented hazy images with diverse haze conditions, enhancing the performance of existing deep dehazing models and reducing the cost of training data collection. The contributions of this work are summarized as follows:

- PANet is a hybrid "physics-guided + data-driven" network that estimates key haze parameters from hazy images: pixel-wise haze density and atmospheric light. It then performs parametric augmentation to generate additional haze patterns, boosting the performance of dehazing models. With physics-based guidance, its lightweight data-driven modules can be effectively trained on a small real-world dataset, leveraging the explainability and efficiency of physics models while minimizing the need for extensive training data.

- PANet operates in two distinct modes for parametric augmentation. In the "learning mode", the haze parameters estimated by HPM are passed directly to PHM without modification, maintaining consistency in the haze-parameter-haze mapping cycle. This minimizes the discrepancy between the augmented data and real-world training data, ensuring more accurate and reliable augmentation. In the augmentation mode, PANet modifies the haze parameter maps to generate additional, previously unseen haze patterns in a physically explainable way, increasing the diversity of training data and improving model generalization.

- Extensive experimental results demonstrate the high efficacy of PANet in boosting state-of-the-art dehazing models on four real-world image dehazing datasets. Cross-dataset evaluations also validate the generalizability of PANet.

## 2 RELATED WORK

### 2.1 IMAGE DEHAZING

Image dehazing techniques have achieved remarkable progress with the fast growth of CNNs. Specifically, to effectively extract haze-related features, several studies resorted to attention-based methods using CNNs. For example, Liu et al. (2019) proposed a multi-scale attention-based network that utilizes channel-wise attention for feature fusion. Qin et al. (2020) proposed a feature fusion attention network with cascaded channel-attention and pixel-attention modules. Deng et al. (2020) proposed a haze-aware representation distillation module to distill haze-related features through instance normalization. Fu et al. (2021) utilized discrete wavelet transform with a generative adversarial network (GAN) to preserve high-frequency knowledge in the feature space. Cui et al. (2023) proposed an efficient image restoration network that contains a dual-domain selection mechanism to emphasize important regions for restoration.

Recently, motivated by Transformers' powerful ability to model long-range dependencies among features, several studies have devised transformer-based models for image dehazing. Song et al.

(2023) utilized window-based attention (Liu et al., 2021) to design a vision transformer for image dehazing. Guo et al. (2022) proposed a hybrid architecture that integrates CNN and Transformer with a transmission-aware 3D position embedding to improve image dehazing performance. Although these methods successfully improve image dehazing performance through elaborate model designs, they primarily rely on synthetic hazy datasets, which may lead to a performance decrease when handling real-world hazy images. Instead of concentrating solely on architectural designs to improve dehazing performance, our goal is to design a haze augmentation method applicable across various dehazing models and improve dehazing performance in real-world scenarios.

## 2.2 Hazy Image Augmentation

Besides improving dehazing performance through architectural innovations, some studies have explored hazy image augmentation strategies to enhance dehazing models. For instance, the method in (Wu et al., 2023) incorporates brightness adjustments, color bias, and Gaussian noise into a physical scattering model to simulate adverse light conditions in real-world scenarios. However, this approach modifies additional factors rather than leveraging the inductive bias of real-world hazy images, which are usually non-homogeneous with high opacity. To generate diverse hazy images with real-world characteristics, Yang et al. (2022) proposed a rehazing model incorporating depth and haze density with CycleGAN (Zhu et al., 2017). By globally sampling haze density, they can generate additional hazy images as a data augmentation operation. However, GAN-based architectures often encounter challenges such as unstable training process (Gulrajani et al., 2017; Mao et al., 2017), model collapse (Akash et al., 2017; Mao et al., 2019), and uncontrollable outputs (Kowalski et al., 2020; Shoshan et al., 2021), which restricts the diversity and usability of the generated images. Furthermore, their method only allows global haze density adjustment, making it unsuitable for real-world hazy images that typically exhibit non-uniformity with high opacity. Chen et al. (2024) propose a test-time adaptation strategy by generating visual prompts to simulate the hazy distribution of the testing set. However, the generated visual prompts often exhibit patch-wise artifacts that deviate significantly from real-world haze distributions. In contrast, our PANet leverages the inherent inductive biases of real-world haze to augment realistic hazy images within a physically explainable framework. PANet enables pixel-wise adjustments of haze conditions, allowing for generating non-homogeneous haze with varying densities and spatial distributions. This significantly enhances the diversity of hazy images, resulting in substantial improvements in the performance of dehazing models across several real-world hazy image datasets

## 3 Proposed Method

### 3.1 Overview

In real-world scenarios, haze is often non-homogeneous and exhibits varying degrees of opacity. To capture these characteristics, PANet is designed to augment photo-realistic hazy images with diverse haze patterns for individual hazy/clean training pairs. This augmentation strategy enhances the diversity of training data, significantly improving the performance of image dehazing models in real-world applications. Figure 2 presents the block diagram of PANet, a cyclic Haze-Parameter-Haze mapping framework comprising a Haze-to-Parameter Mapper (HPM) and a Parameter-to-Haze Mapper (PHM). Given a hazy/clean image pair, the HPM maps the hazy image into a learned parametric space, characterizing real-world haze conditions with two pixel-wise maps: haze density and atmospheric light. In the parametric space, these maps are either kept fixed (learning mode) or modified (augmentation mode). The augmented parameters are then applied to generate an initial hazy image from the clean image using the physical scattering model (Eq. 1.) To address the inherent inaccuracies of haze parameter estimation caused by the ill-posed nature of the problem, the Data-driven Haze Refiner (DHR) $N_{\text{DHR}}(\cdot)$ is employed to refine the initial hazy image, ensuring more realistic and accurate haze simulation.

By resampling the pixel-wise haze parameters during the parametric augmentation process, we can generate additional hazy images beyond those in the original training set. These newly augmented images feature diverse and physically explainable haze conditions not previously seen in the training data. This significantly enriches the training set, leading to improved performance of existing dehazing models in real-world scenarios. Next, we will introduce the core components of PANet, including HPM, PHM, and the parametric augmentation process.

Figure 2: Block diagram of PANet. PANet utilizes a cyclic haze-parameter-haze mapping framework consisting of a Haze-to-Parameter Mapper (HPM) followed by a Parameter-to-Haze Mapper (PHM). Besides the hazy images in the original training set (green boxes), PANet can augment additional hazy images with various haze conditions unseen in the training set (yellow boxes).

## 3.2 HAZE-TO-PARAMETER MAPPER (HPM)

HPM operates in two steps: parametric mapping and augmentation. First, it maps hazy images into a learned parametric space that captures haze characteristics using two physically interpretable parameters: haze density and atmospheric light. Then, it augments these parameter maps to generate diverse haze patterns. HPM comprises an encoder and two decoders, as shown in Figure 3. Given hazy image $I_H(z) \in \mathbb{R}^{H \times W \times 3}$, where $z$ is the pixel index, the encoder $E_{\text{haze}}(\cdot)$ extracts haze-specific features of $I_H(z)$. Next, the Haze Density Decoder $D_{\text{HD}}(\cdot)$ and the Atmospheric Light Decoder $D_{\text{AL}}(\cdot)$ are used to estimate the pixel-wise haze density map $\beta_{\text{est}}(z) \in \mathbb{R}^{H \times W \times 3}$ and atmospheric light map $A_{\text{est}}(z) \in \mathbb{R}^{H \times W \times 1}$, respectively, as

$$\beta_{\text{est}}(z) = D_{\text{HD}}(E_{\text{haze}}(I_H(z))), \tag{2}$$

$$A_{\text{est}}(z) = D_{\text{AL}}(E_{\text{haze}}(I_H(z))). \tag{3}$$

To derive $d(z)$ in Eq. (1), we choose RA-Depth (Mu et al., 2022) as the pre-trained depth estimator $\Psi(\cdot)$ similar to RIDCP (Wu et al., 2023). Besides, to bridge the domain gap with the pre-trained depth estimator, we further use a Depth Refinement Module (DRM) $d_{\text{ref}}$ to refine the depth map as

$$d(z) = d_{\text{ref}}(\Psi(I_C(z))), \tag{4}$$

where the architecture of $d_{\text{ref}}(\cdot)$ is similar to HPM but with only one decoder.

To accurately estimate $\beta_{\text{est}}(z)$, $A_{\text{est}}(z)$, and $d(z)$ in the "training mode" of HPM, we keep the estimated parameters unchanged and use them to generate a reconstructed hazy image from the input clean image using Physical scattering Model (PSM) and DHR in PHM. The fidelity between the input hazy image and its reconstructed version is then measured to assess the accuracy of the estimated parameters, providing supervision for training the learnable modules.

## 3.3 PARAMETER-TO-HAZE MAPPER (PHM)

After retrieving the haze parameters and scene depth, we further utilize PHM to map the haze parameters back to real hazy images. Specifically, based on the estimated $\beta_{\text{est}}(z)$ and $A_{\text{est}}(z)$, we subsequently translate the input clean image $I_C(z)$ to the initial hazy image $O_{\text{ini}}(z) \in \mathbb{R}^{H \times W \times 3}$ using the following Physical Scattering Model:

$$O_{\text{ini}}(z) = I_C(z)t(z) + A_{\text{est}}(z)(1 - t(z)), \tag{5}$$

$$t(z) = e^{-\beta_{\text{est}}(z)d(z)}, \tag{6}$$

where $d(z) \in \mathbb{R}^{H \times W \times 1}$ denotes the depth map estimated from the clean image $I_C(z)$.

By using the Physical Scattering Model, we can generate initial hazy images $O_{\text{ini}}(z)$ from the clean image $I_C(z)$. This model provides physical meanings for the haze parameter estimated by HPM.

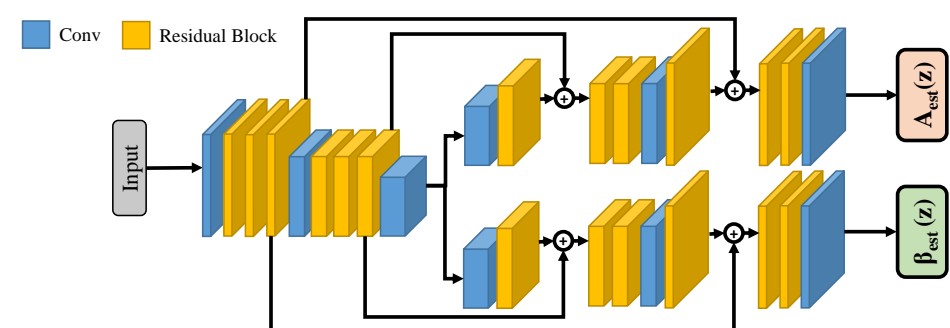

Figure 3: Haze-to-Parameter Mapper (HPM) consists of a shared encoder followed by two parallel parameter decoders to estimate the haze density map $\beta_{\text{est}}(z)$ and atmospheric light map $A_{\text{est}}(z)$.

However, since a scene in a dense, non-homogeneous haze is usually substantially occluded or distorted, retrieving pixel-wise haze parameters from a hazy image is highly ill-posed, making the model-based initial hazy images inaccurate and distorted (*e.g.*, incorrect color tone and unrealistic transparency) (Gui et al., 2023; Zhang et al., 2021; Chen et al., 2021b). Therefore, we propose a Data-driven Haze Refiner (DHR) $N_{\text{DHR}}(\cdot)$ to further refine the initial hazy images to mitigate the inaccuracy. To this end, we concatenate $O_{\text{ini}}(z)$ with its corresponding clean image $I_C(z)$ and feed them to $N_{\text{DHR}}(\cdot)$ to get the real hazy image $O_{\text{final}}(z)$ as

$$O_{\text{final}}(z) = N_{\text{DHR}}(\text{Concate}(O_{\text{ini}}(z), I_C(z))), \tag{7}$$

where $N_{\text{DHR}}(\cdot)$ has a similar architecture to HPM but with only one decoder.

With the cyclic haze-parameter-haze mapping involving HPM and PHM, PANet can successfully project hazy images into a parameter space and then generate additional hazy images by pixel-wisely augmenting the haze density $\beta_{\text{est}}(z)$ and atmospheric light $A_{\text{est}}(z)$ with physically-explainable haze conditions unseen in the training set, as elaborated in the haze augmentation process.

### 3.4 LOSS FUNCTION

We choose Charbonnier loss (Lai et al., 2017) $\mathcal{L}_{\text{char}}$ and perceptual loss (Johnson et al., 2016) $\mathcal{L}_{\text{perc}}$ for optimizing PANet as follows:

$$\begin{aligned}
\mathcal{L}_{\text{total}} = \ &\mathcal{L}_{\text{char}}(O_{\text{ini}}(z), I_H(z)) + \mathcal{L}_{\text{char}}(O_{\text{final}}(z), I_H(z)) \\
&+ \lambda \mathcal{L}_{\text{perc}}(O_{\text{ini}}(z), I_H(z)) + \lambda \mathcal{L}_{\text{perc}}(O_{\text{final}}(z), I_H(z)),
\end{aligned} \tag{8}$$

where $O_{\text{ini}}(z)$ denotes the model-generated initial hazy image, $O_{\text{final}}(z)$ denotes the refined hazy images, $I_H(z)$ denotes the ground-truth hazy image, and $\lambda$ is a weight empirically set to $\lambda = 10^{-6}$.

### 3.5 PARAMETRIC AUGMENTATION OF HAZE

To generate new hazy images unseen in the training set, given a pair of hazy $I_H(z)$ and clean $I_C(z)$ images, in the "augmentation mode" of HPM, we modify the estimated haze density map $\beta_{\text{est}}(z) \in \mathbb{R}^{H \times W \times 3}$ and atmospheric light map $A_{\text{est}}(z) \in \mathbb{R}^{H \times W \times 1}$ to obtain their new versions: $\beta'(z)$ and $A'(z)$. The two new maps are then used to generate a new hazy image by using PHM. Specifically, we can alter haze density $\beta_{\text{est}}(z)$ by a scaling factor $\alpha$ to generate $\beta'(z)$ as

$$\beta'(z) = \alpha \cdot \beta_{\text{est}}(z), \tag{9}$$

For example, two new hazy images with $\alpha = 0.5$ and $\alpha = 2$ are illustrated in Figs. 4(A) and 4(B), respectively. As illustrated in Figure 4(C), we can also reverse the location of haze patterns by altering the atmospheric light map $A_{\text{est}}(z)$ as

$$A'(z) = 1 - A_{\text{est}}(z), \tag{10}$$

where $A_{\text{est}}(z)$ ranges in $[0, 1]$. In this case, for those reverse regions that do not contain haze in the original hazy image, we sample $\beta'(z)$ to be in $[0.6, 1.25]$, the range of $\beta_{\text{est}}$ in the whole training set. Moreover, we can linearly interpolate $A_{\text{est}}(z)$ and $1 - A_{\text{est}}(z)$ to generate $A'(z)$ as

$$A'(z) = \min(\gamma A_{\text{est}}(z) + \eta(1 - A_{\text{est}}(z)), 1), \tag{11}$$

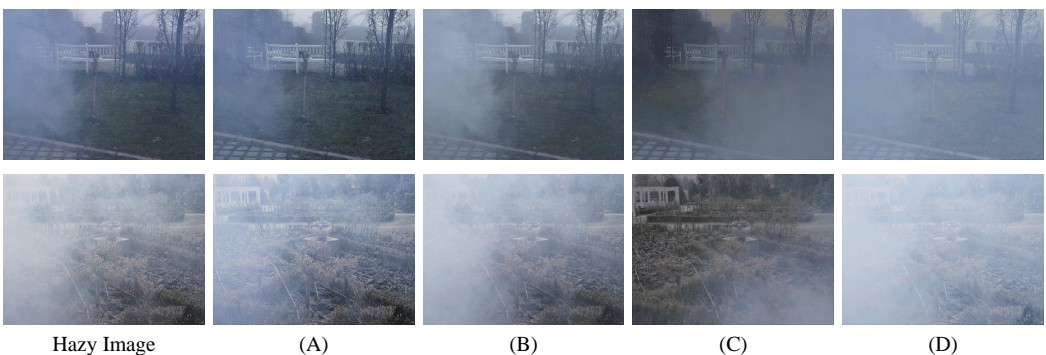

Figure 4: Visuals of hazy images generated by PANet. Given a hazy image, we can decrease or amplify its haze density by 2, as shown in (A) and (B). In addition, we can reverse its haze location or generate a complex hazy image, as shown in (C) and (D)

where $\gamma$ and $\eta$ denote the weights for $A_{\text{est}}(z)$ and $1 - A_{\text{est}}(z)$, respectively. As shown in Figure 4(D), we generate diverse hazy images for each hazy-clean training pair by modifying the haze density and spatial distribution through the haze augmentation process. Unlike traditional augmentation techniques applied in the image domain, our parametric space augmentation enables precise, physically interpretable control over haze patterns, allowing for more realistic and diverse haze conditions.

## 4 EXPERIMENTS

### 4.1 IMPLEMENTATION DETAILS

**PANet.** We train PANet on NH-Haze20 (Ancuti et al., 2020) consisting of non-homogeneous hazy and clean image pairs captured in real-world outdoor scenes. Following the settings in (Fu et al., 2021; Cui et al., 2023), we use 50 training pairs and 5 testing pairs. During training, we utilize the Adam optimizer with an initial learning rate of $5 \times 10^{-5}$, which is then reduced to $10^{-7}$ using a cosine annealing schedule. PANet is trained for 270 epochs with a batch size of 2. To augment the data, we apply random cropping of $256 \times 256$ patches along with random rotations and flips.

**Dehazing Models.** We adopt three state-of-the-art (SOTA) dehazing models, including DW-GAN (Fu et al., 2021), DeHamer (Guo et al., 2022), and FocalNet (Cui et al., 2023), to evaluate the effectiveness of PANet. To make a fair comparison, we utilize the 50 training pairs of NH-Haze20 to train the SOTA dehazing models as their baseline following the default training setting in their methods. We then utilize PANet to generate 400 additional training pairs, 8 times larger than the original 50 training pairs. We use the augmented training set with 450 pairs to retrain the SOTA dehazing models to obtain the PANet-enhanced version of their baseline. We evaluate the performances of the above dehazing models on four real-world hazy image datasets, including NH-Haze20 (Ancuti et al., 2020) test set, NH-Haze21 (Ancuti et al., 2021) dataset, O-Haze (Ancuti et al., 2018b) test set, and I-Haze (Ancuti et al., 2018a) test set. Specifically, NH-Haze20 test set contains 5 testing pairs with non-homogeneous haze. Since NH-Haze21 does not provide a test set, we use its training set that consists of 25 pairs captured in non-homogeneous hazy environments for evaluation. In contrast, O-Haze and I-Haze test sets contain 5 outdoor and 5 indoor testing pairs with homogeneous haze, respectively. Besides, we further utilize RTTS (Li et al., 2019a) and Fattal's (Fattal, 2014) datasets that collected 4322 and 31 hazy images in real-world environments without ground truth clean images to evaluate the performance.

### 4.2 PERFORMANCE EVALUATIONS

**Quantitative Performance Comparison.** Table 1 compares the dehazing performances of three baselines and their PANet-enhanced versions, where "Baseline" and "+PANet" denote the dehazing performances without and with PANet, respectively. As shown in Table 1, the PANet-augmented dataset significantly improves the average PSNR performances of the three dehazing models, including DW-GAN (Fu et al., 2021), DeHamer (Guo et al., 2022), and FocalNet (Cui et al., 2023) by 0.50 dB, 0.75 dB, 1.57 dB, and 0.48 dB on NH-Haze20 (Ancuti et al., 2020), NH-Haze21 (Ancuti et al.,

Table 1: Quantitative performances of different dehazing methods on NH-Haze20 test set, NH-Haze21 dataset, O-Haze test set, and I-Haze test set. "Baseline" and "+PANet" represent the dehazing performance without and with PANet, respectively.

| Model | | NH-Haze20 | | NH-Haze21 | | O-Haze | | I-Haze | |
|---|---|---|---|---|---|---|---|---|---|
| | | PSNR (dB) | SSIM | PSNR (dB) | SSIM | PSNR (dB) | SSIM | PSNR (dB) | SSIM |
| DW-GAN | Baseline | 21.50 | 0.697 | 18.10 | **0.726** | 18.44 | 0.574 | 14.88 | 0.403 |
| | +PANet | **21.84 (+0.34)** | **0.704** | **18.42 (+0.32)** | 0.708 | **20.15 (+1.71)** | **0.634** | **15.47 (+0.59)** | **0.508** |
| DeHamer | Baseline | 20.01 | 0.649 | 16.49 | 0.612 | 20.01 | 0.600 | 15.49 | 0.463 |
| | +PANet | **20.73 (+0.72)** | **0.650** | **17.05 (+0.56)** | **0.627** | **20.64 (+0.63)** | **0.650** | **16.22 (+0.73)** | **0.563** |
| FocalNet | Baseline | 20.31 | 0.646 | 16.51 | 0.632 | 18.28 | 0.622 | 15.29 | **0.417** |
| | +PANet | **20.76 (+0.45)** | **0.682** | **17.87 (+1.36)** | **0.700** | **20.64 (+2.36)** | **0.639** | **15.41 (+0.12)** | 0.374 |
| Average Gain | | **+0.50** | **+0.015** | **+0.75** | **+0.022** | **+1.57** | **+0.042** | **+0.48** | **+0.054** |

Table 2: Quantitative performances of different dehazing methods on RTTS and Fattal's dataset.

| Method | RTTS | | | | | Fattal's | | | | |
|---|---|---|---|---|---|---|---|---|---|---|
| | NIQE ↓ | PIQE ↓ | BRISQUE ↓ | MUSIQ ↑ | PAQ2PIQ ↑ | NIQE ↓ | PIQE ↓ | BRISQUE ↓ | MUSIQ ↑ | PAQ2PIQ ↑ |
| DW-GAN (Baseline) | 3.46 | 37.10 | 25.23 | **58.09** | **69.74** | 3.48 | 32.31 | **25.62** | **67.56** | **75.62** |
| DW-GAN (+PANet) | **3.06** | **34.00** | **23.99** | 57.81 | 68.94 | **3.18** | **29.07** | 26.42 | 66.63 | 75.19 |
| Dehamer (Baseline) | 3.44 | 54.92 | 32.74 | **55.70** | **69.43** | 3.21 | 34.86 | 26.95 | 65.73 | 73.61 |
| Dehamer (+PANet) | **3.33** | **50.83** | **32.10** | 55.64 | 69.06 | 3.52 | **29.60** | **23.76** | **66.35** | **74.37** |
| FocalNet (Baseline) | 3.38 | 55.83 | 33.58 | 56.00 | 67.96 | **3.07** | 34.99 | **24.22** | **66.55** | 73.89 |
| FocalNet (+PANet) | 3.38 | **52.57** | **32.02** | **56.13** | **68.9** | 3.11 | **33.68** | 26.54 | 66.32 | **74.19** |

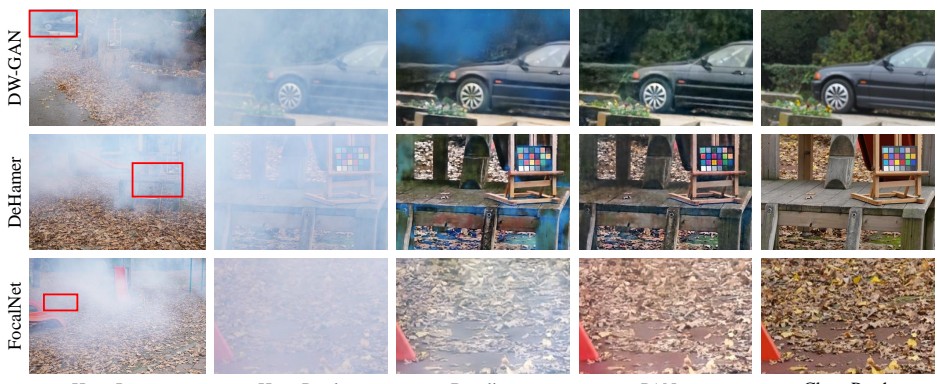

Figure 5: Qualitative performance comparison on NH-Haze21 (Ancuti et al., 2021) dataset.

2021), O-Haze (Ancuti et al., 2018b) and, I-Haze (Ancuti et al., 2018a), respectively. In Table 2, we demonstrate the effectiveness of PANet on RTTS (Li et al., 2019a) and Fattal's Fattal (2014) datasets. Since both RTTS and Fattal's does not provide ground truth clean images, we utilize five no-reference quality metrics, NIQE (Mittal et al., 2013), PIQE (N et al., 2015), BRISQUE (Mittal et al., 2012), MUSIQ (Ke et al., 2021), and PAQ2PIQ (Ying et al., 2020), to evaluate performance. PANet consistently improves these three dehezing models. These evaluation results demonstrate that PANet can effectively help boost the performances of deep dehazing models under various haze conditions.

**Qualitative Performance Comparison.** We demonstrate some dehazed images of the dehazing models with or without using PANet in 5 and 6. Figure 5 visualizes some dehazed results on NH-Haze21. Compared to their baselines, PANet-enhanced models achieve significant visual quality improvements by removing unwanted hazy artifacts or correcting color distortions. In Figure 6, we visualize dehazed results on RTTS. Again, the PANet-enhanced dataset can also significantly boost the performances of state-of-the-art models under haze conditions in various real-world scenarios. These visuals show that PANet is effective in augmenting both homogeneous and non-homogeneous hazy images in real-world scenarios. We demonstrate more visualization results on NH-Haze20, I-Haze, O-Haze, RTTS, and Fattal's datasets in the section Appendix.

### 4.3 ABLATION STUDIES

In the ablation studies, we analyze the impact of PANet on the dehazing performance of FocalNet on NH-Haze20 test set, where"Baseline" denotes the PSNR performance of FocalNet trained on NH-Haze20 training set without using the additional training pairs augmented by PANet.

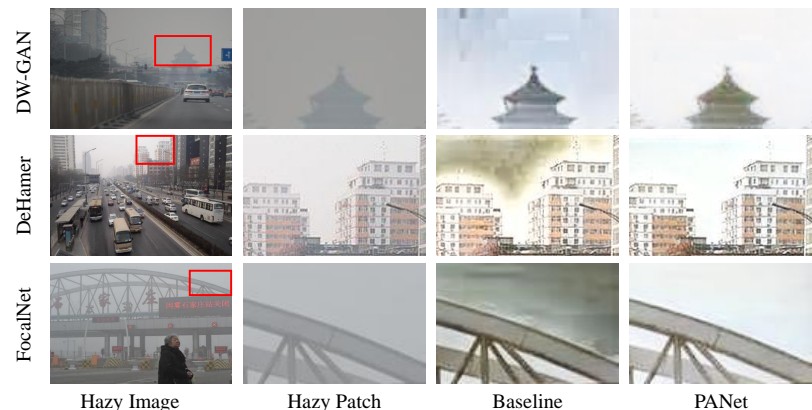

Figure 6: Qualitative performance comparison on RTTS (Li et al., 2019a) dataset.

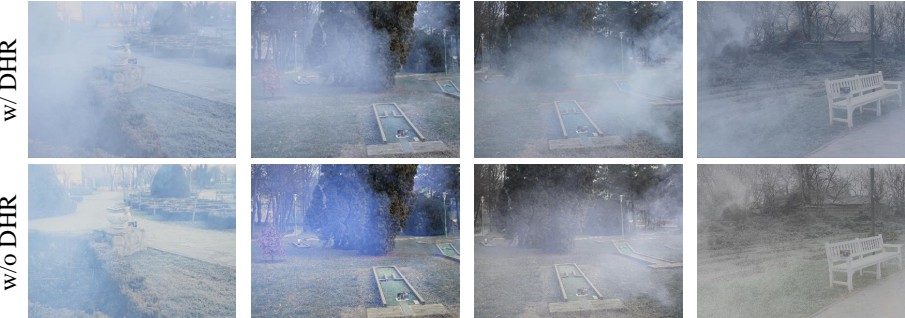

Figure 7: Effect of the Data-driven Haze Refiner (DHR) in PANet. First row: Hazy images generated by PANet with DHR. Second row: Hazy images generated by PANet without DHR.

Table 3: Effect of the Data-driven Haze Refiner (DHR) $N_{\text{DHR}}(\cdot)$ in PSNR in PANet.

| FocalNet | Baseline | w/o DHR | with DHR |
|----------|----------|---------|----------|
| NH-Haze20 | 20.31 dB | 20.22 dB | **20.76 dB** |

**Effect of Data-driven Haze Refiner (DHR).** To verify the effectiveness of DHR, as shown in Table 3, we compare the dehazing performance enhanced by PANet with and without DHR, denoted "w/o DHR" and "w/ DHR", respectively. The results show that PANet without DHR, which generates hazy images by solely using the physical scattering model, cannot improve the Baseline, due to the inaccuracy of estimated haze parameters. Besides, the visuals in Figure 7 show that, without using DHR, the generated hazy images contain unrealistic color tones and transparency compared to the ones using DHR, demonstrating the importance of DHR in PANet.

**Effect of depth estimation network and Depth Refinement Module (DRM).** The pre-trained depth estimator may suffer from a domain gap when addressing unseen clean images, thereby degrading the accuracy of the physical scattering model (Othman & Abdulla, 2022; Lou et al., 2023). However, estimating the transmission map from a single hazy image is a highly ill-posed problem. Therefore, we use a pre-trained depth estimator combined with DRM to refine the initial depth map. In Table 4, we compare PANet with two versions of PANet: PANet without the depth estimator and DRM and PANet without DRM, denoted "w/o depth and DRM" and "w/ DRM", respectively. The results demonstrate that the final version of PANet achieves the best performance.

**Effect of the Number of Augmented Images.** To assess the impact of the PANet-augmented training set size on dehazing performance, we generate various numbers of augmented pairs to improve the baseline model trained on 50 original pairs. Table 5 presents the results using additional 200, 400, and 600 augmented pairs, representing $400\%$, $800\%$, and $1,200\%$ increases in dataset size, respectively. The findings indicate that dehazing performance improves as more augmented pairs are added but tends to plateau when the number reaches 600. Based on this, we opt to augment 400 additional pairs, striking an optimal balance between training time and performance gains

Table 4: Effectiveness of the Depth Refinement Module (DRM) in PSNR in PANet.

| FocalNet | Baseline | +PANet (w/o depth and DRM) | +PANet (w/o DRM) | +PANet |
|---|---|---|---|---|
| NH-Haze20 | 20.31 | 20.39 | 20.31 | **20.76** |

Table 5: Effect of the dehazing performance in PSNR versus the number of augmented pairs by PANet, where the original training set contains 50 training pairs.

| FocalNet | Baseline (0%) | 200 (400%) | 400 (800%) | 600 (1200%) |
|---|---|---|---|---|
| NH-Haze20 | 20.31 | 20.51 | 20.76 | **20.83** |

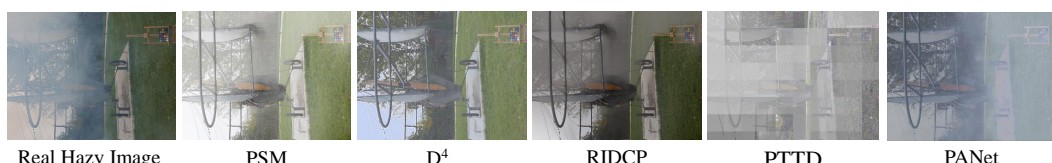

Real Hazy Image     PSM     $D^4$     RIDCP     PTTD     PANet

Figure 8: Hazy images generated by different augmentation methods, including Physical Scattering Model (PSM), $D^4$ (Yang et al., 2022), RIDCP (Wu et al., 2023), PTTD (Chen et al., 2024), and PANet.

Table 6: Comparison regarding dehazing performance in PSNR with different augmentation methods, including Physical Scattering Model (PSM), $D^4$ (Yang et al., 2022), RIDCP (Wu et al., 2023), PTTD (Chen et al., 2024), and PANet.

| FocalNet | Baseline | w/ PSM | w/ $D^4$ | w/ RIDCP | w/ PTTD | w/ PANet |
|---|---|---|---|---|---|---|
| NH-Haze20 | 20.31 | 20.41 | 20.26 | 20.57 | 19.41 | **20.76** |

**Comparison with different augmentation methods.** Finally, we compare PANet with different haze augmentation methods, including uniform haze method: Physical Scattering Model (PSM), GAN-based method: $D^4$ (Yang et al., 2022), PTTD (Chen et al., 2024), and RIDCP (Wu et al., 2023), where we use these methods to augment the NH-Haze20 training set to improve FocalNet and evaluate on the NH-Haze20 testing set. As shown in Table 6, our PANet achieves the best performance compared to other augmentation methods. In addition, we compare the hazy images generated by these methods in Fig. 8. Among these methods, PSM and RIDCP (Wu et al., 2023) solely rely on the physical scattering models to generate hazy images. Although RIDCP can alter the brightness and color bias, they both cannot generate non-homogeneous hazy images. $D^4$ (Yang et al., 2022) applies a cycle-GAN-based architecture to generate hazy images. However, the lack of robustness of GAN increases the difficulty of generating realistic hazy images. In addition, GAN-based methods cannot pixel-wisely control haze conditions to generate diverse hazy images. The visual prompt generated by PTTD (Chen et al., 2024) exhibits patch-wise artifacts that deviate significantly from real-world haze distributions. In contrast, our PANet is a robust network through the physics-guided learning strategy and can pixel-wisely alter hazy conditions to generate diverse non-homogeneous hazy images.

**Limitations and Future Works** In this work, we develop PANet by leveraging the inductive biases inherent in real-world hazy images, such as haze density and atmospheric light. As a result, PANet is specifically designed for the dehazing task at this stage. In the future, we plan to extend the concept of PANet to other tasks, such as desmoking (Jin et al., 2022), deraining, desnowing, to further benefit a broader range of image restoration applications.

## 5 CONCLUSION

We proposed a Parametric Augmentation Network (PANet) to generate diverse non-homogeneous hazy images, enhancing the performance of dehazing models in real-world scenarios. PANet consists of a Haze-to-Parameter Mapper, which projects hazy images into a parametric space, and a Parameter-to-Haze Mapper, which maps the augmented parameters back into hazy images. By modifying the estimated haze parameter maps, PANet generates hazy images with various haze patterns unseen in the training set. This enables the creation of diverse training pairs, improving the robustness of dehazing models. Extensive experiments demonstrate that PANet effectively boosts the performance of three SOTA dehazing models across five real-world hazy image benchmarks.

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

# A APPENDIX

We provide additional results to further validate the effectiveness of PANet. First, we give more performance evaluations between PANet and the GAN-based augmentation method: $D^4$ (Yang et al., 2022). Second, we conduct qualitative comparisons between PANet and existing haze augmentation methods. Third, we present visualization results of feature maps in PANet. Fourth, we provide a user study of PANet on the RTTS (Li et al., 2019a) dataset and demonstrate additional qualitative results of dehazed images enhanced by PANet in real-world hazy scenes. Lastly, we demonstrate several augmented hazy images generated by PANet, along with their corresponding atmospheric light and haze density maps.

## A.1 COMPARISON BETWEEN PANET AND $D^4$

In our method, we utilize a physics-guided learning strategy to optimize PANet, offering several advantages for PANet to generate realistic hazy images. First, PANet can map hazy images into a haze parameter space involving haze density and atmospheric light so that their haze conditions can easily adjusted in a physically meaningful manner by resampling the parameter space to generate diverse hazy images. Second, with the physical scattering model to generate initial hazy images as guidance, PANet can be realized with a relatively simple model that only contains 3M parameters and requires 23 GFLOPs, with an inference time of 25 ms for 256×256 images. Third, PANet does not rely on a large amount of training data. We can effectively optimize PANet using only 50 pairs of hazy/clean images. In contrast, $D^4$ is on top of a CycleGAN-based architecture, which lacks the controllable ability to generate diverse hazy images. In addition, $D^4$ cannot be adequately optimized with only 50 pairs of hazy/clean images since GAN-based methods require much more training data to learn robust statistic distributions. Furthermore, $D^4$ contains 11M parameters, which is 8M more than that of PANet.

## A.2 QUALITATIVE COMPARISONS BETWEEN PANET AND EXISTING HAZE AUGMENTATION METHODS.

In Figure 9, we adopt FocalNet as the dehazing model and present further qualitative comparisons between PANet and existing haze augmentation methods on NH-Haze21 datasest. PANet consistently outperforms competing methods by effectively removing unwanted hazy patterns.

## A.3 VISUALIZATION RESULTS OF FEATURE MAPS IN PANET.

In Figure 10, we present feature maps predicted by PANet, including the depth map, atmospheric light map, depth map refined by DRM, haze density map, and transmission map. Notably, DRM refines the depth map generated by the pre-trained depth estimator, producing a refined depth map that is more suitable for haze generation.

## A.4 USER STUDY ON THE RTTS DATASET

We have conducted a double-blind subjective user study to further evaluate the performance. For each dehazing model, we randomly selected 20 dehazed images, with or without using PANet, for comparison. In total, we selected 20×3 = 60 pairs in the user study. Next, we recruited 20 participants we did not know beforehand and asked them to indicate their preference regarding the dehazing quality. The result, with a p-value of $2.54\mathrm{e}^{-17}$ (less than 0.05), shows that PANet-enhanced results received 63% of the preference votes, demonstrating the effectiveness of PANet on RTTS.

## A.5 DEHAZING RESULTS ON REAL-WORLD HAZY SCENES

To demonstrate the effectiveness of PANet in real-world hazy scenarios, we qualitatively compare the dehazing performances of dehazing models, including DW-GAN (Fu et al., 2021), De-

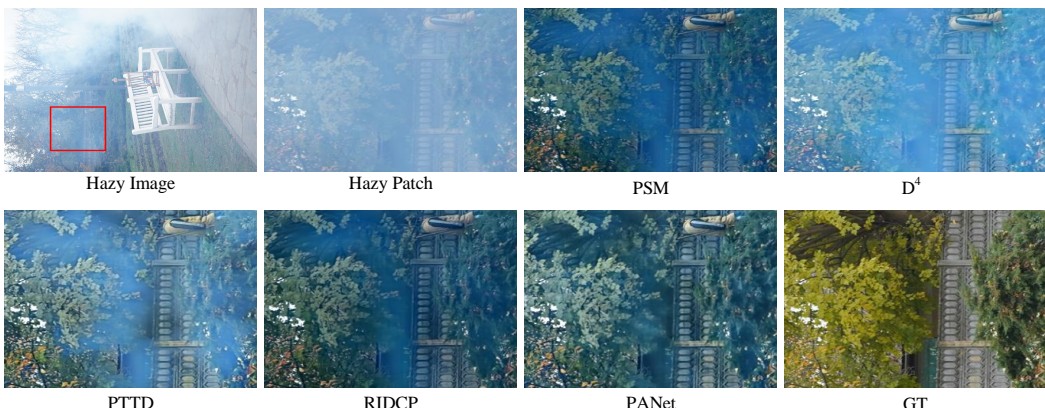

Figure 9: Qualitative comparisons between PANet and existing haze augmentation methods, including PSM, D4 (Yang et al., 2022), RIDCP (Wu et al., 2023), PTTD (Chen et al., 2024), and PANet.

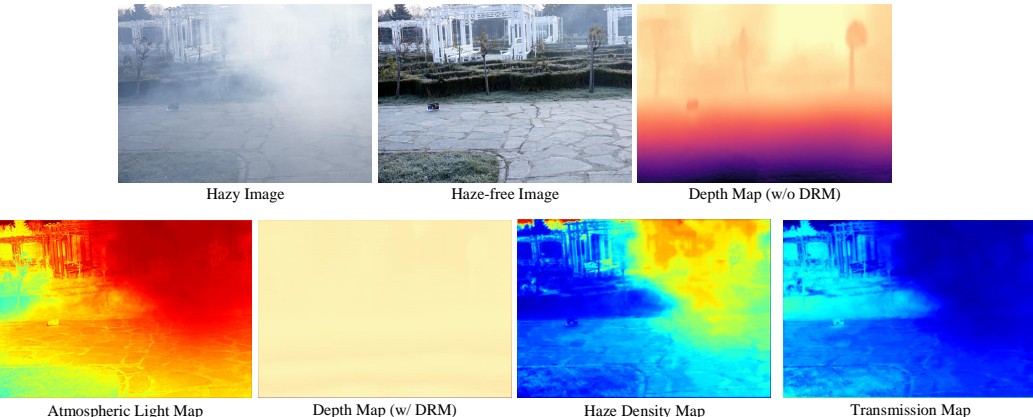

Figure 10: Visualization results of depth map, atmospheric light map, depth map refined by DRM, haze density map, and transmission map.

Hamer (Guo et al., 2022), and FocalNet (Cui et al., 2023), with or without using PANet-augmented data on the NH-Haze20 (Ancuti et al., 2020), I-Haze (Ancuti et al., 2018a), O-Haze (Ancuti et al., 2018b), and RTTS datasets (Li et al., 2019a). We demonstrate the dehazed images on NH-Haze20, I-Haze, and O-Haze in Figures 11, 12, and 13, respectively, where "Baseline" indicates the dehazing models without using PANet-augmented data, and "PANet" represents their PANet-enhanced versions. Additionally, images in RTTS are collected in real-world hazy environments without ground truth reference images. We show the dehazed images on the RTTS with FocalNet in Figures 14 and 15, with DeHamer in Figures 16 and 17, and with DW-GAN in Figures 18 and 19. Moreover, we show the dehazed images on the Fattal's (Fattal, 2014) dataset with FocalNet in Figures 20, with DeHamer in Figures 21, and with DW-GAN in Figures 22.

A.6    VISUALS OF HAZY IMAGES AUGMENTED BY PANET

We then demonstrate several augmented hazy images by PANet and their corresponding haze density and atmospheric light maps in 23 and 24. In the top parts of 23 and 24, we show the original hazy images and their estimated haze density $\beta_{est}(z)$ and atmospheric light $A_{est}(z)$. In the bottom parts of 23 and 24, we show the augmented hazy images and their corresponding haze density $\beta'(z)$ and atmospheric light $A'(z)$. By pixel-wisely resampling the hazy density and atmospheric light, we can generate diverse hazy images unseen in the training set. Since we parameterize various haze conditions into the haze density and atmospheric light to capture the characteristics of haze, the atmospheric light also contains haze-related information that can be used to adjust hazy patterns.

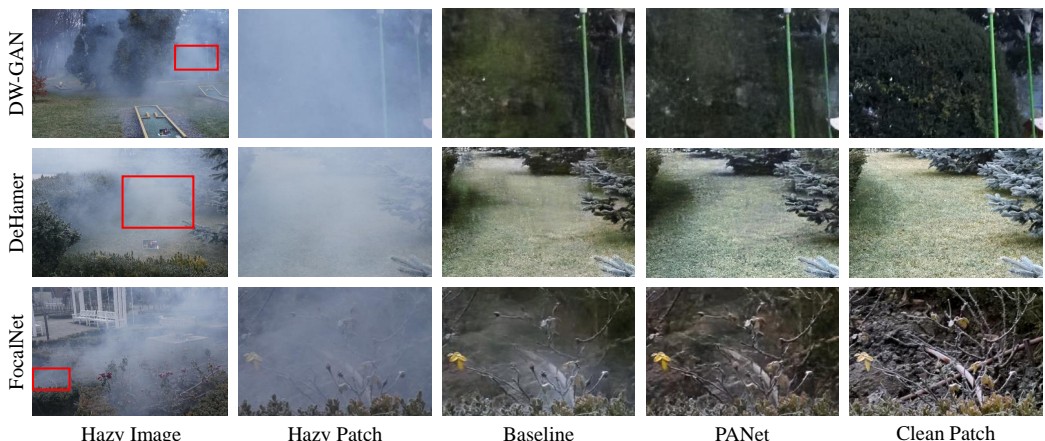

Figure 11: Qualitative performance comparison on NH-Haze20 (Ancuti et al., 2020) dataset.

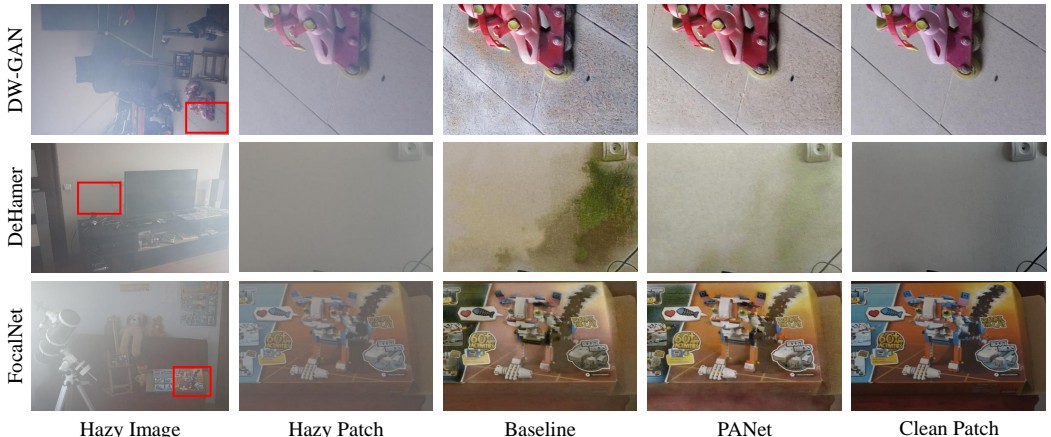

Figure 12: Qualitative performance comparison on I-Haze (Ancuti et al., 2018a) dataset.

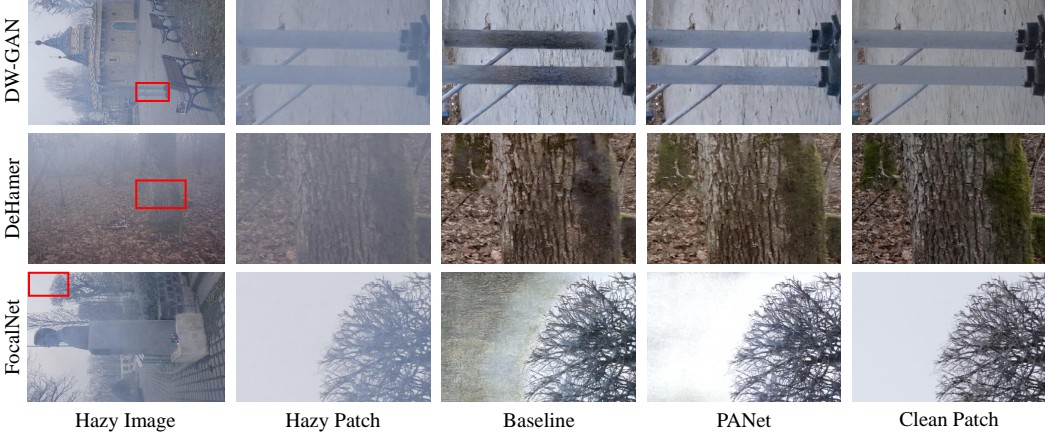

Figure 13: Qualitative performance comparison on O-Haze (Ancuti et al., 2018b) dataset.

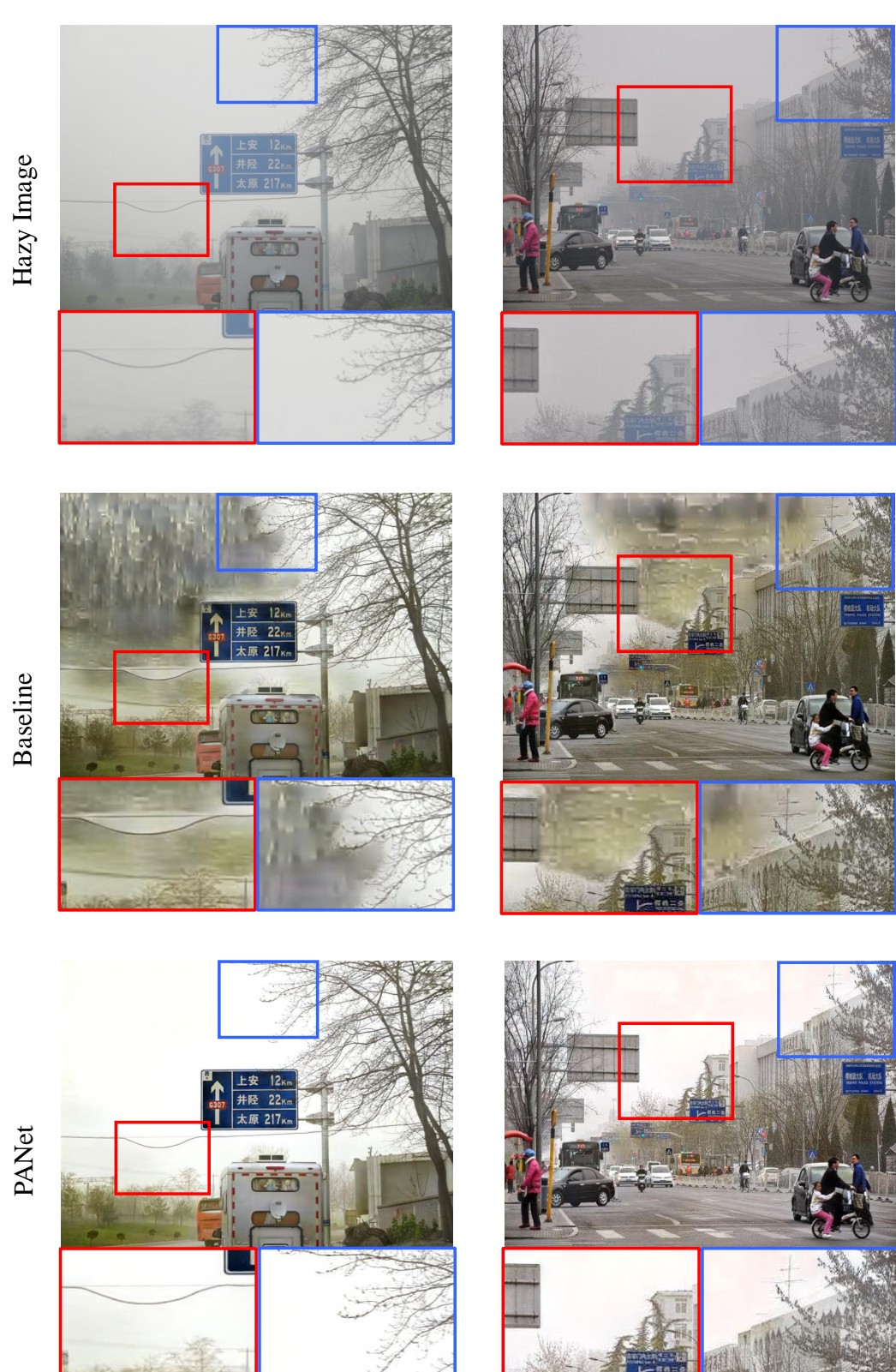

Figure 14: Qualitative results of FocalNet (Cui et al., 2023) on the RTTS (Li et al., 2019a) dataset.

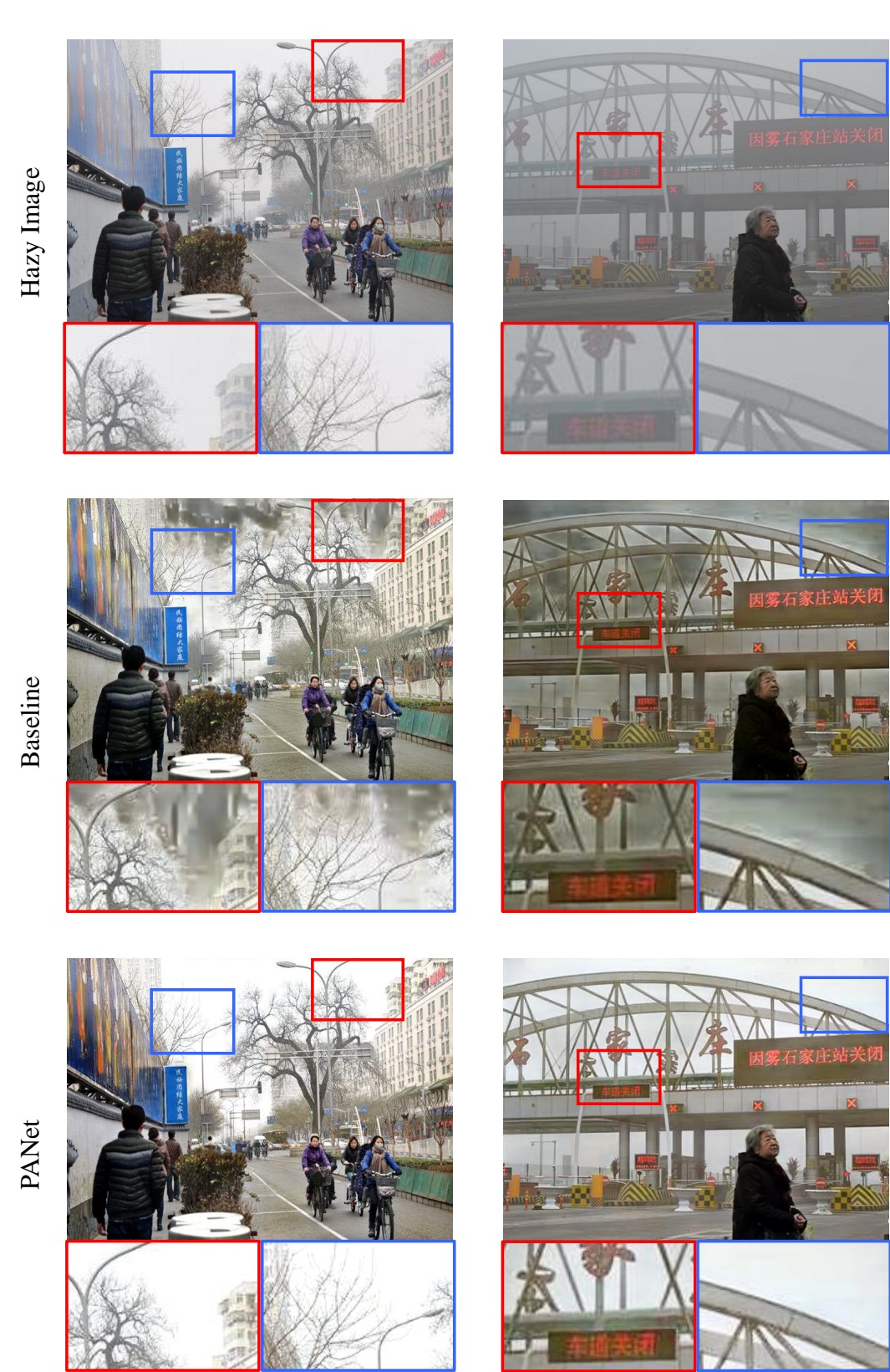

Figure 15: Qualitative results of FocalNet (Cui et al., 2023) on the RTTS (Li et al., 2019a) dataset.

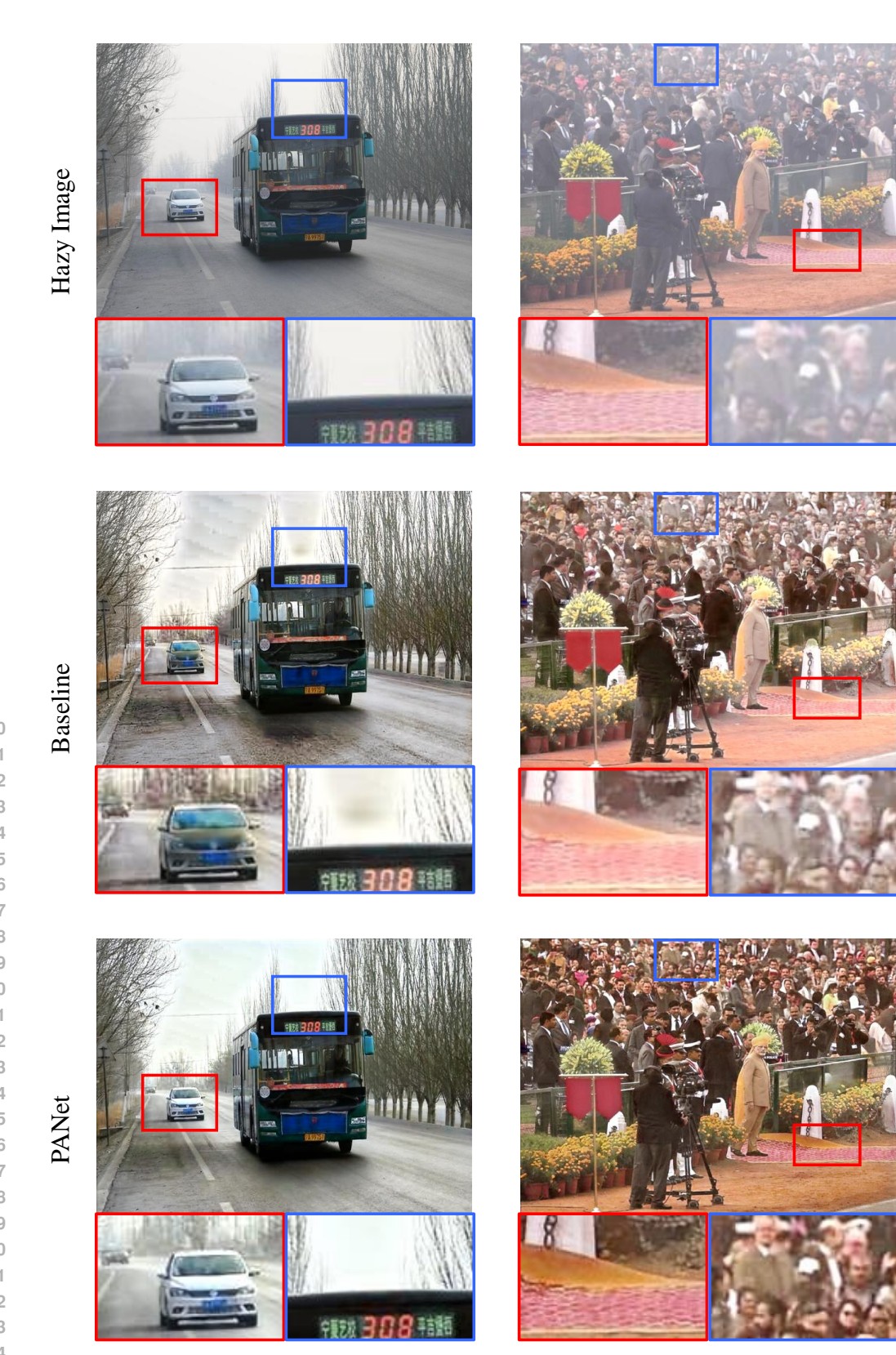

Figure 16: Qualitative results of DeHamer (Guo et al., 2022) on the RTTS (Li et al., 2019a) dataset.

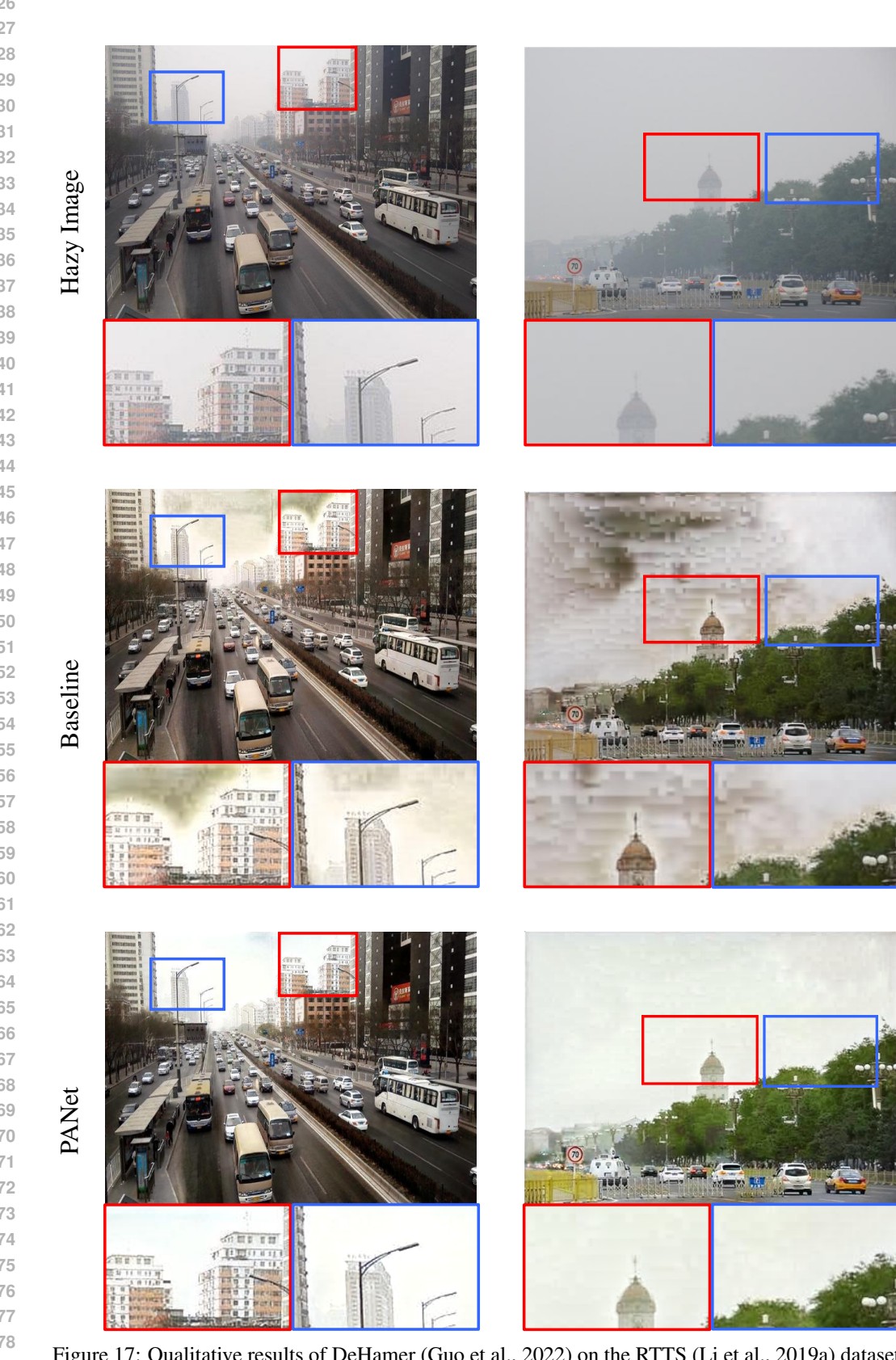

Figure 17: Qualitative results of DeHamer (Guo et al., 2022) on the RTTS (Li et al., 2019a) dataset.

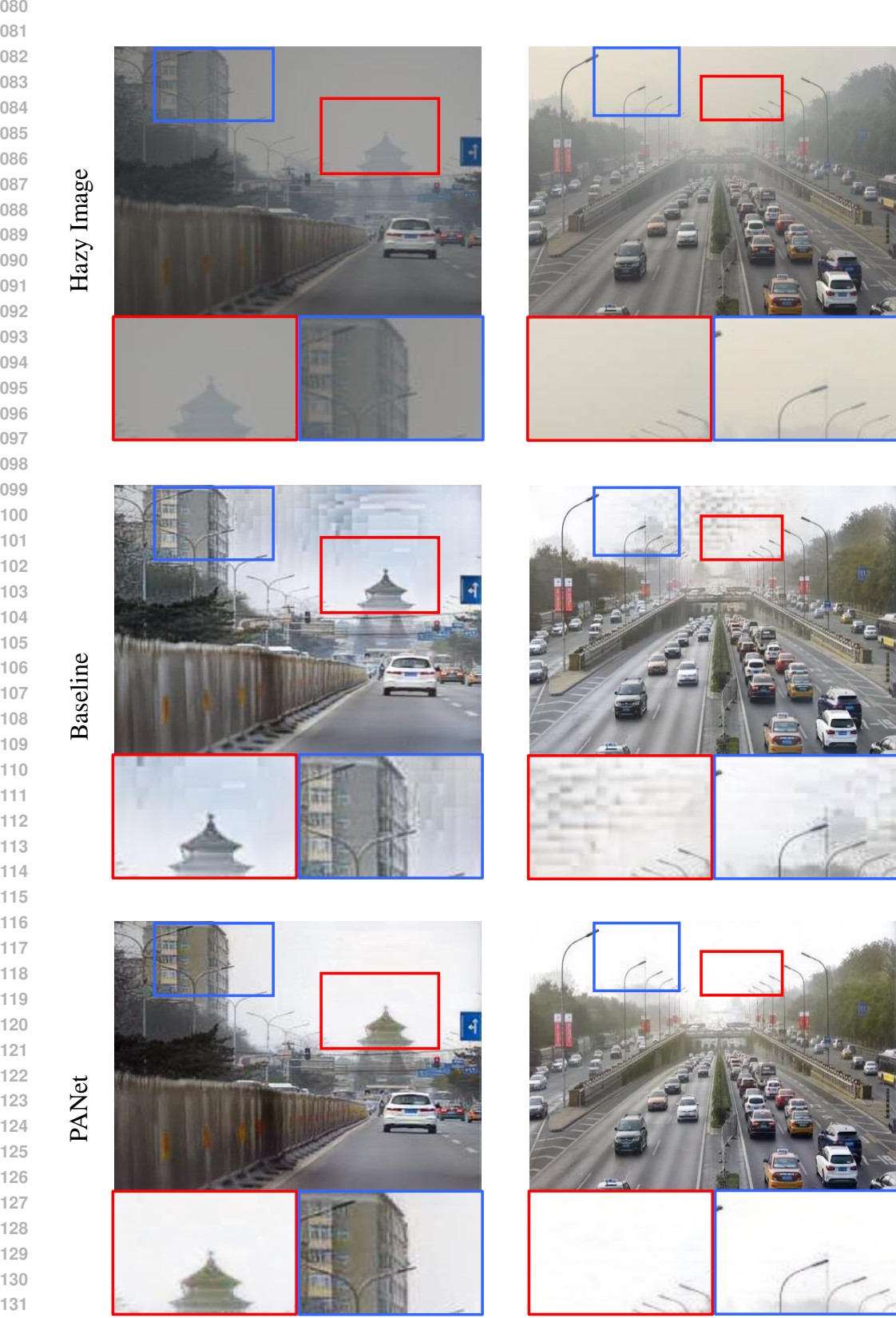

Figure 18: Qualitative results of DW-GAN (Fu et al., 2021) on the RTTS (Li et al., 2019a) dataset.

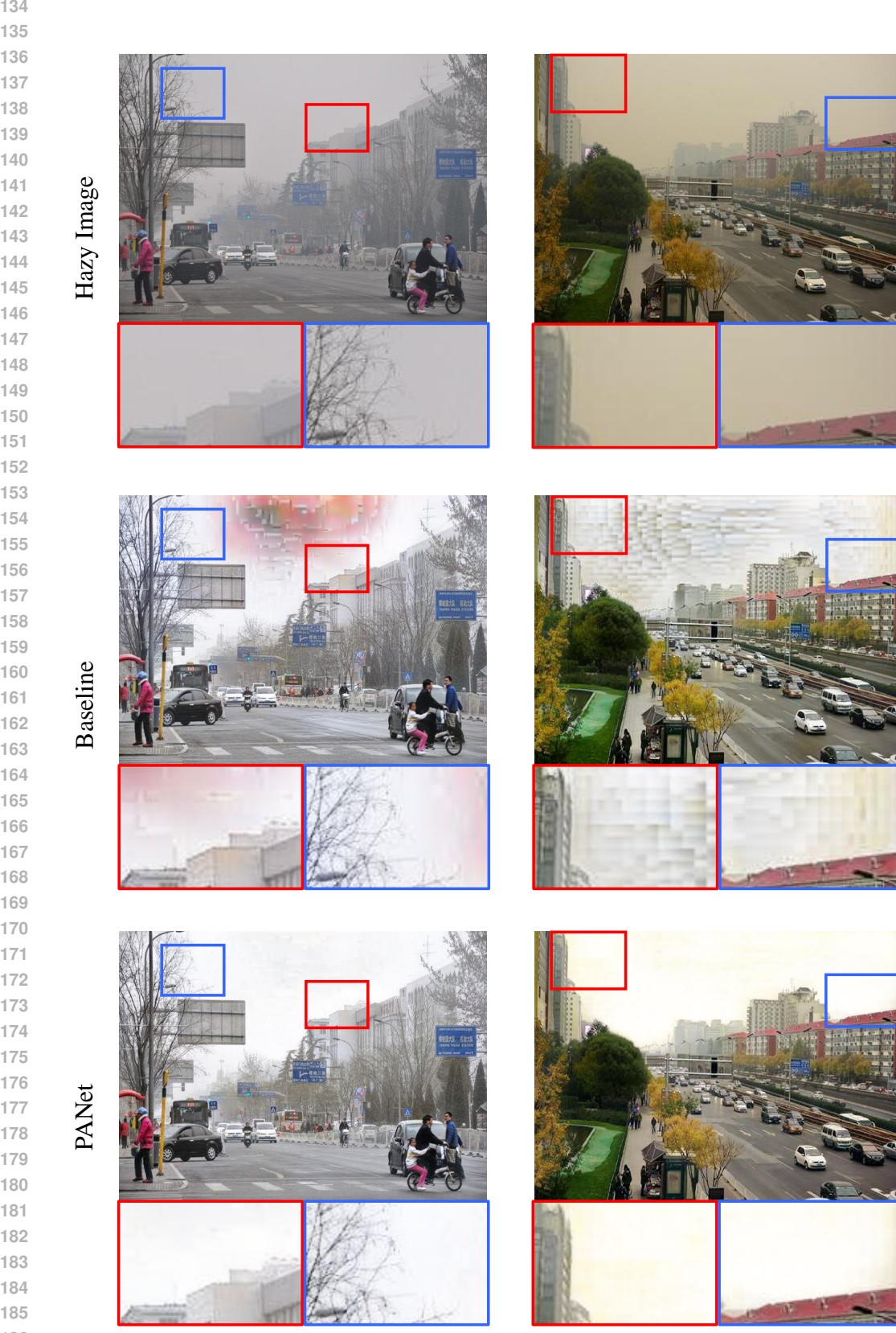

Figure 19: Qualitative results of DW-GAN (Fu et al., 2021) on the RTTS (Li et al., 2019a) dataset.

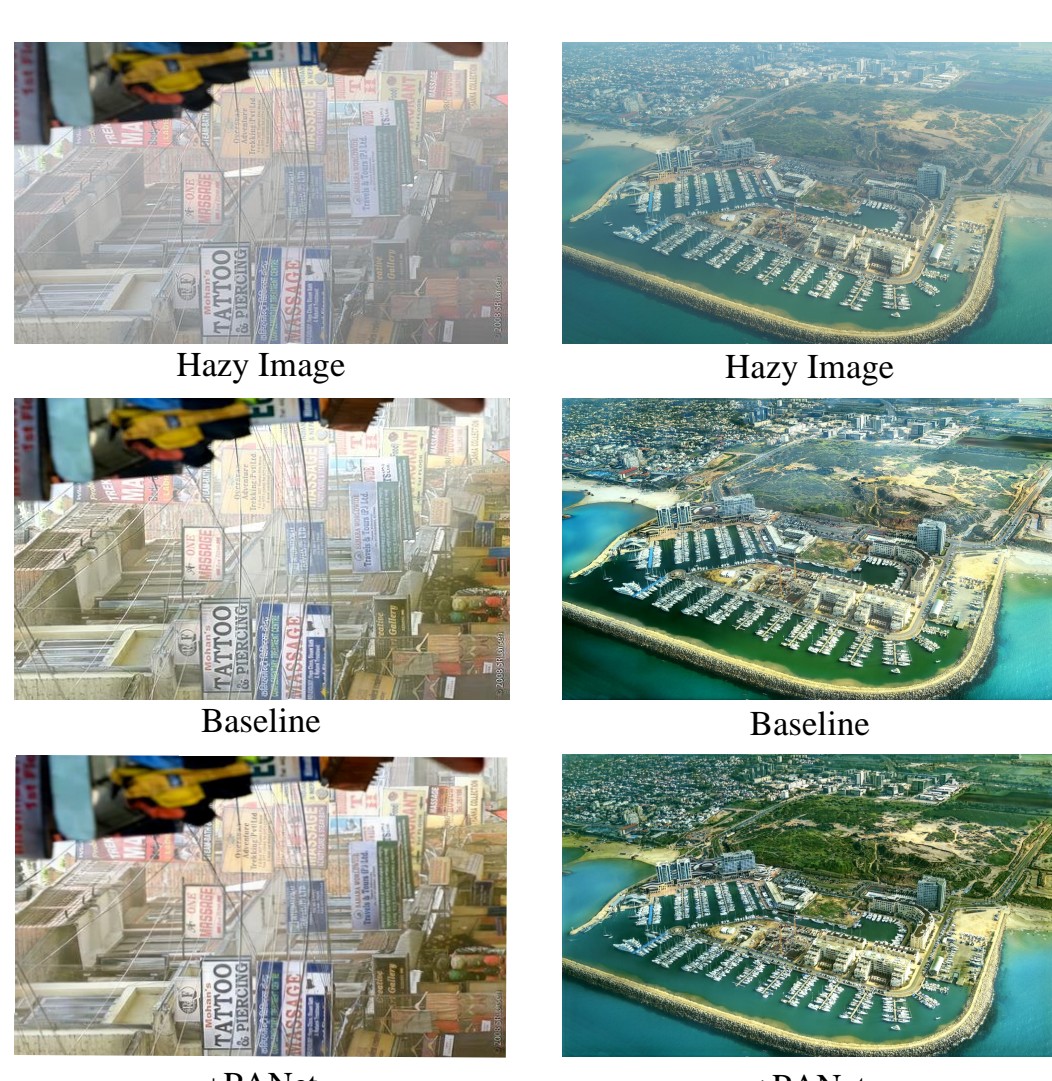

Hazy Image                    Hazy Image

Baseline                      Baseline

+PANet                        +PANet

Figure 20: Qualitative results of FocalNet (Cui et al., 2023) on the Fattal's (Fattal, 2014) dataset.

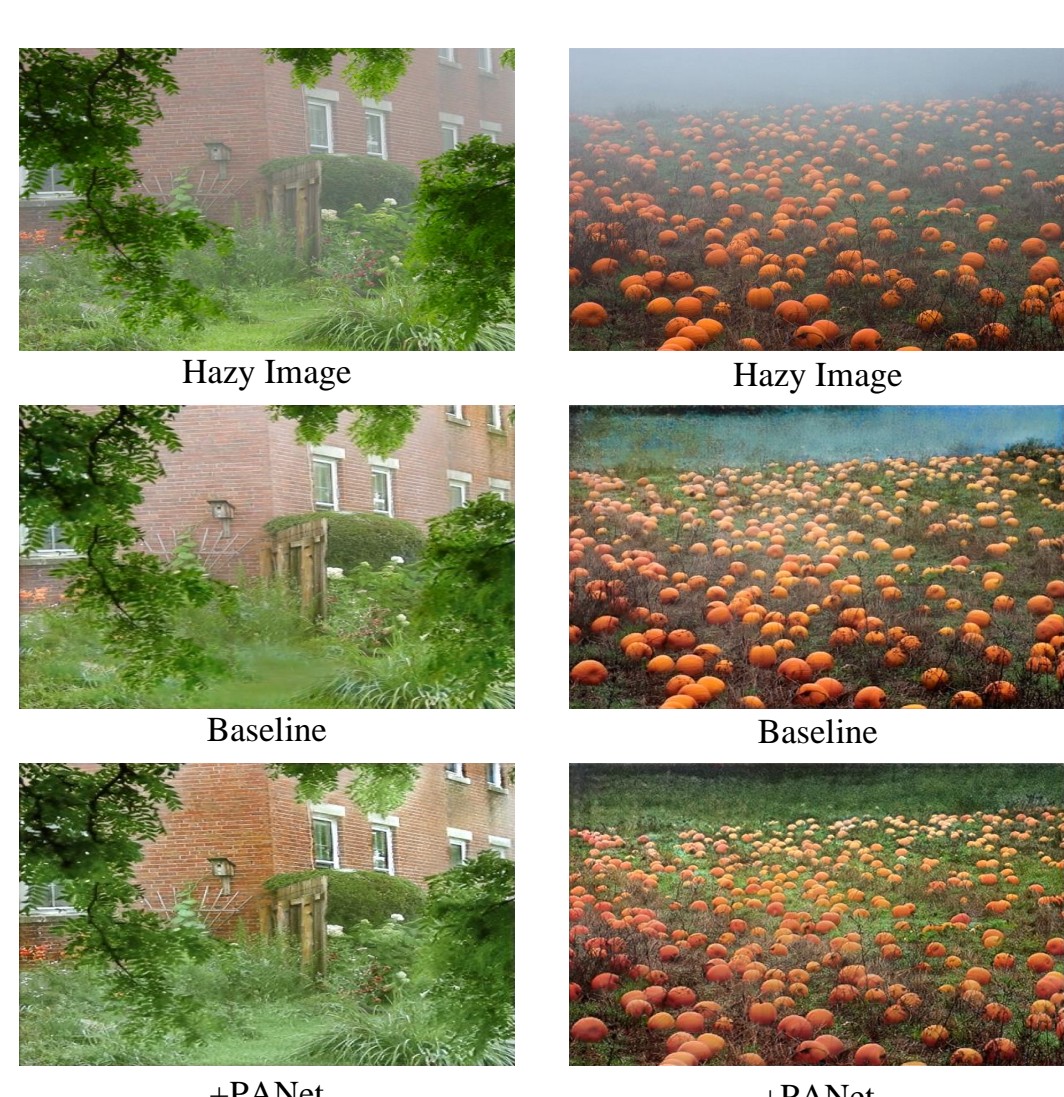

Figure 21: Qualitative results of Dehamer (Guo et al., 2022) on the Fattal's (Fattal, 2014) dataset.

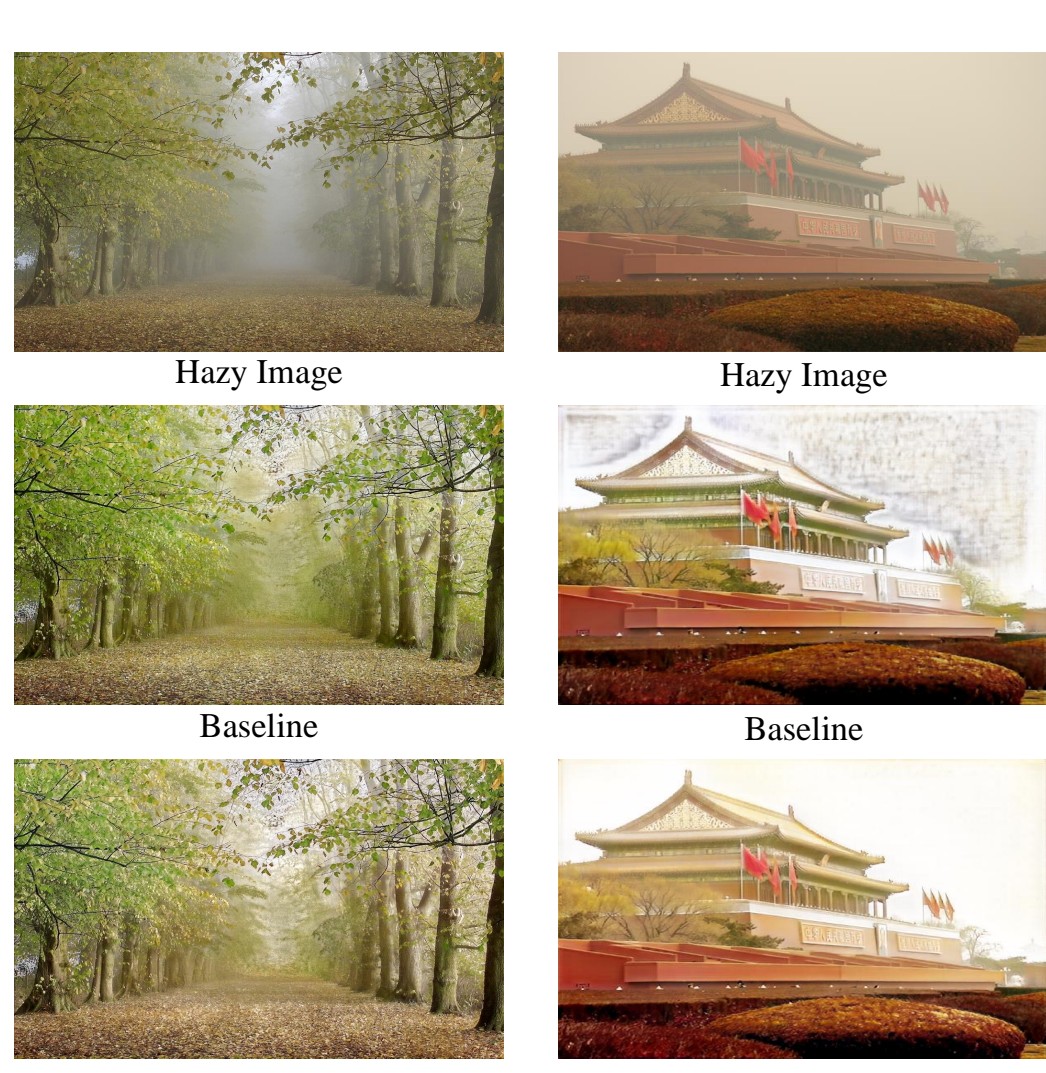

Figure 22: Qualitative results of DW-GAN (Fu et al., 2021) on the Fattal's (Fattal, 2014) dataset.

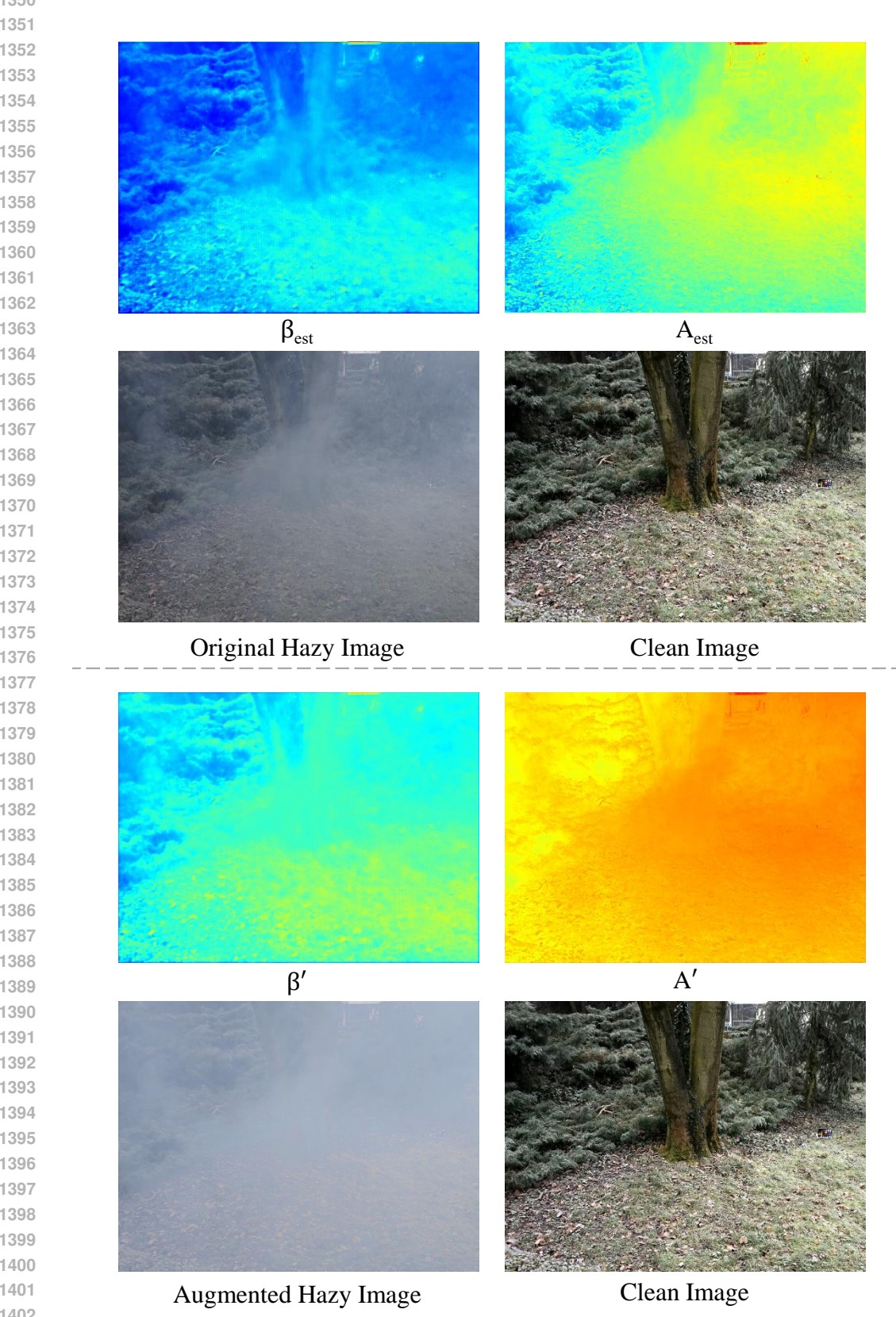

Figure 23: Qualitative results of hazy images generated by PANet.

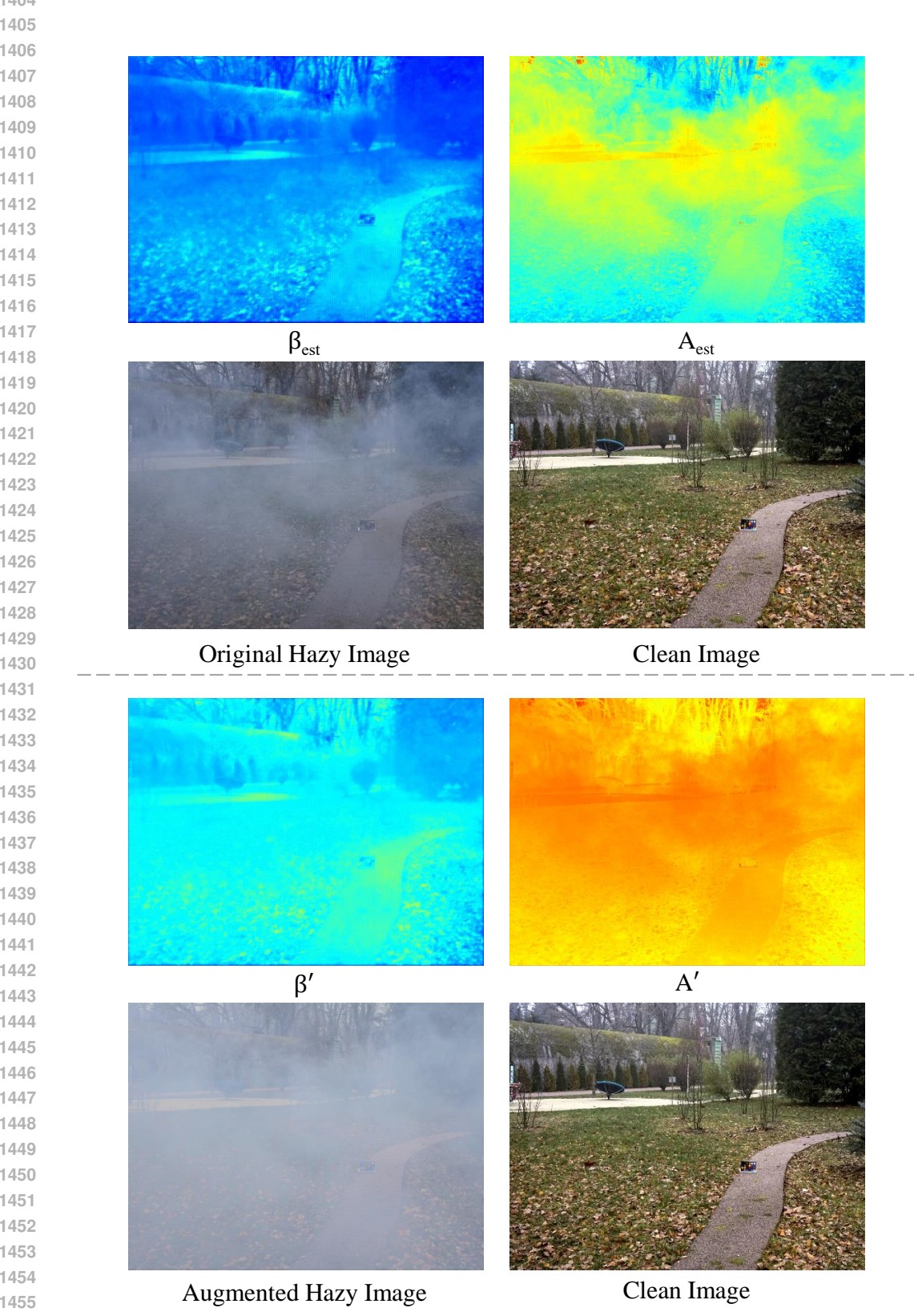

Figure 24: Qualitative results of hazy images generated by PANet.

