# OpenReview forum: "Unleashing the Power of Deep Dehazing Models: A Physics-guided Parametric Augmentation Net for Image Rehazing"
_ICLR.cc/2025/Conference — ICLR 2025 Conference Withdrawn Submission_

### Official Review · Reviewer_BjsZ · 2024-11-01

**Soundness:** 2
**Presentation:** 3
**Contribution:** 2
**Rating:** 5
**Confidence:** 4

**Summary:**

The paper presents a data augmentation pipeline specifically designed for real-world dehazing. This method utilizes a physical scattering model of haze, adjusting model parameters estimated by neural networks. For each real-world training patch, the approach can generate an arbitrary number of new patches with varying haze densities. The authors validate the proposed method by demonstrating its capacity to enhance the real-world dehazing performance of three state-of-the-art dehazing techniques across four datasets.

**Strengths:**

This work possesses several strengths:

- It effectively enhances the richness of real-world dehazing training data, thereby improving the real-world dehazing performance of existing methods.

- It allows users to create an arbitrary number of new patches with varying haze densities.

**Weaknesses:**

However, the work exhibits several shortcomings:

- The contributions appear insufficient. Although it shows slight improvements over [1], the core concept remains quite similar. The claimed distinctions, such as the GAN structure and global haze adjustment, can be categorized as engineering problems rather than scientific advancements. The proposed method, which employs simple ResBlocks and pixel-wise haze adjustment, may be viewed as incremental.

- The proposed method has not been compared with existing data augmentation techniques for dehazing via DeHamer and DW-GAN on the other three datasets. While such augmentation methods could enhance existing dehazing approaches, it is essential to assess the generalizability of these improvements. However, the work neglects to compare its method against these data augmentation techniques for more general cases.

- The manuscript requires revision. For instance, L237 references Eq. ??. Additionally, Figures 2 and 3 are nearly identical, differing only in minor content details.

- The work evaluates visual performance on RTTS, which is designed for haze detection. It would be beneficial to conduct experiments on dehazing in the context of object detection to assess how real-world dehazing after data augmentation impacts downstream object detection tasks.

- According to Table 5, a larger number of augmented data pairs improves dehazing performance, yet the authors stop at 600. It would be useful to evaluate the convergence of dehazing performance relative to the number of augmentations.

[1] Self-augmented unpaired image dehazing via density and depth decomposition, CVPR 2022.
[2] RIDCP: Revitalizing real image dehazing via high-quality codebook priors, CVPR 2023.

**Questions:**

The questions have been outlined in the section on weaknesses. Considering the aforementioned strengths and weaknesses, I would recommend a borderline reject, with the potential for reconsideration if the listed issues are adequately addressed.

---

> ### Author Response · Authors · 2024-11-23
> **Author Feedback 1**
>
> * **Contributions of PANet.**
>
> In this paper, we observe that effective data augmentation methods for real-world image dehazing have been underexplored. To address this gap, we propose PANet, an effective and robust haze augmentation method. D4 [1], a GAN-based approach, relies on unpaired synthetic hazy images to extract global haze coefficients. However, GANs often suffer from unstable training processes, lack of robustness, and difficulty in controlling pixel-wise haze conditions. Moreover, the limited training pairs provided by existing real-world dehazing datasets make it challenging to effectively optimize GAN-based methods.
>
> In contrast, PANet is a physics-guided parametric augmentation network that automatically disentangles pixel-wise haze coefficients without relying on haze coefficient labels or GAN structures. We introduce an innovative approach by projecting haze coefficients into a parameter space, implicitly characterizing factors such as atmospheric light and haze density. This allows us to sample diverse haze conditions to augment hazy images not present in the training set. We believe that this implicit and physically guided process has the potential to advance the field of real-world image dehazing and inspire new research directions.
>
> * **Comparisons between PANet and existing data augmentation methods via Dehamer and DW-GAN.**
>
> Thank you for the suggestion. As the experiments take time to finish, we will provide additional comparisons with Dehamer and DW-GAN within a few days.
>
> * **Revision of the manuscript in L237 and Figures 2 and 3.**
>
> We have merged Figures 2 and 3 and fixed typos and reference errors in the updated paper.
>
> * **Effectiveness of PANet on RTTS for object detection.**
>
> Thank you for the suggestion. We will evaluate the impact of PANet-enhanced dehazing on the performance of downstream object detection on RTTS and provide the result soon.
>
> * **Experiments on a larger number of augmented data pairs.**
>
> Thank you for the suggestion. We will generate additional augmented images and provide experiments on a larger number of augmented data pairs within a few days.
>
> [1] Self-augmented unpaired image dehazing via density and depth decomposition. In CVPR, 2022.

---

> ### Author Response · Authors · 2024-11-25
>
> Dear BjsZ,
>
> We have made every effort to address your concerns and sincerely hope our responses meet your expectations. We would greatly appreciate your feedback on our replies to help us further enhance the manuscript.

---

> > ### Comment · Reviewer_BjsZ · 2024-11-26
> > **Looking forward to the additional results**
> >
> > Dear Authors,
> >
> > To facilitate a more comprehensive evaluation of the proposed method, especially the data augmentation technique, I recommend including an ablation study on data augmentation, conducting detection on the RTTS (though this is not necessary), and incorporating an additional setting with augmented data pairs. As these aspects are still forthcoming, I am unable to adjust my rating at this time. I look forward to reviewing the updated version. Preliminary results would suffice given the time constraints.
> >
> > Best regards

---

> > > ### Author Response · Authors · 2024-11-26
> > > **Comparisons between PANet and existing data augmentation methods via Dehamer and DW-GAN.**
> > >
> > > **Table A. Cross-dataset comparisons of dehazing performance of haze augmentation methods, including RIDCP, D4, PTTD, and PANet.**
> > >
> > > | FocalNet | NH-Haze20 | NH-Haze21 | O-Haze | I-Haze | Average |
> > > |----------|-----------|-----------|--------|--------|---------|
> > > | Baseline | 20.31     | 16.51     | 18.28  | 15.29  | 17.60   |
> > > | +RIDCP   | 20.57     | 17.02     | 19.11  | **16.06** | 18.19   |
> > > | +D4      | 20.26     | 14.57     | 19.48  | 15.58  | 17.47   |
> > > | +PTTD    | 19.41     | 16.06     | 18.31  | 14.95  | 17.18   |
> > > | +PANet   | **20.76** | **17.87** | **20.64** | 15.41  | **18.67** |
> > >
> > > | Dehamer | NH-Haze20 | NH-Haze21 | O-Haze | I-Haze | Average |
> > > |----------|-----------|-----------|--------|--------|---------|
> > > | Baseline | 20.01     | 16.49    | 20.01  | 15.49  | 18.00   |
> > > | +RIDCP   | 20.62     | 16.35     | 19.86  | **16.54** | 18.34   |
> > > | +D4      | 19.83    | 15.15     | 19.46  | 14.99  | 17.36   |
> > > | +PTTD    | 18.88    |  16.31    | **21.15**  | 15.47  | 17.95   |
> > > | +PANet   | **20.73** | **17.05** | 20.64 | 16.22  | **18.66** |
> > >
> > > | DW-GAN| NH-Haze20 | NH-Haze21 | O-Haze | I-Haze | Average |
> > > |----------|-----------|-----------|--------|--------|---------|
> > > | Baseline | 21.50    | 18.10    | 18.44  | 14.88  | 18.23   |
> > > | +RIDCP   | 21.73     | 18.36     | 18.98  | 14.05 | 18.28   |
> > > | +D4      | 21.34    | 17.87     | 19.69  | 14.94  | 18.46   |
> > > | +PTTD    | 12.87    |  12.80    | 10.16  | 14.28  | 12.53   |
> > > | +PANet   | **21.84** | **18.42** | **20.15** | **15.47**  | **18.97** |
> > >
> > > To further demonstrate the effectiveness of PANet, we perform cross-dataset validation comparisons, as shown in Table A. In this evaluation, three dehazing models, including FocalNet, Dehamer, and DW-GAN, are trained on the baseline NH-Haze20 dataset and four expanded versions of NH-Haze20, augmented using PANet, RIDCP, D4, and PTTD [1]. These models are then tested on four different datasets to assess their generalizability. The results indicate that the PANet-enhanced model consistently outperforms competing models across nearly all datasets, highlighting that PANet-augmented data effectively captures more diverse haze distributions.
> > >
> > > We will very soon provide other experiments, including experiments on a larger number of augmented data pairs and effectiveness of PANet on RTTS for object detection.
> > >
> > > [1] Prompt-Based Test-Time Real Image Dehazing: A Novel Pipeline. In ECCV 2024.

---

> ### Author Response · Authors · 2024-12-02
> **Detection results on the RTTS dataset and additional setting with augmented data pairs**
>
> | FocalNet   | Baseline (0%) | 200 (400%) | 400 (800%) | 600 (1200%) | 800 (1400%) |
> |------------|---------------|------------|------------|-------------|-------------|
> | **NH-Haze20** | 20.31         | 20.51      | 20.76      | **20.83**   | 20.37       |
>
> (1) We have conducted additional experiments to evaluate the convergence of dehazing performance relative to the number of augmentations. Additionally, we performed experiments using a larger number of augmented data pairs ($800\%$). The results show that PANet converges when using $600$ augmented data.
>
>
> | RTTS (AP$_{50}$)   | DW-GAN  | Dehamer | FocalNet  |
> |---------------------|---------|---------|-----------|
> | **Hazy Inputs**     | **0.7190** | **0.7190** | 0.7190    |
> | **Baseline**        | 0.7101  | 0.7164  | 0.7210    |
> | **Ours**            | 0.7116  | 0.7150  | **0.7220** |
>
> (2) Per your suggestion, We have conducted experiments to evaluate the impact of dehazing on downstream object detection using YoloV8 as the detection model on RTTS. However, we observed that the ground-truth bounding boxes in the RTTS dataset are incomplete, leading the detection model to predict results that appear more accurate than the ground truth. As a result, the AP scores can not accurately reflect the dehazing performance. Due to time constraints, we have not included visualization results of object detection in the Appendix. We will provide these results after publication.

---

### Official Review · Reviewer_digo · 2024-11-02

**Soundness:** 2
**Presentation:** 3
**Contribution:** 2
**Rating:** 5
**Confidence:** 5

**Summary:**

This paper introduces a Physics-guided Parametric Augmentation Network (PANet) to address the domain gaps between synthetic and real-world haze data. PANet is designed to generate real haze images along with their corresponding clean pairs. It consists of two components: the Haze-to-Parameter Mapper (HPM), which projects hazy images into a parametric space, and the Parameter-to-Haze Mapper (PHM), which maps haze parameters back to hazy images.

**Strengths:**

1. The paper tries to address a meaningful problem: bridging domain gaps between synthetic and real-world data.
2. The structure and presentation of the paper are clear and well-organized.

**Weaknesses:**

1. The authors do not account for the idealized assumptions of the physical scattering model, which may lead to inaccuracies in haze removal.
2. The importance of using real-world natural haze images, beyond the non-homogeneous haze created by fog machines, is overlooked.
3. The proposed approach heavily relies on existing datasets, which lack diversity (environment, light condition, etc.).

**Questions:**

1. The physical scattering model is an idealized approximation and may not accurately represent real-world haze, which can still cause domain gaps. Have the authors considered this limitation, and are there any solutions to mitigate it?
2. The training dataset NH-Haze20 is generated using a fog machine, which creates domain discrepancies between these images and those with natural real-world haze. Even with the proposed augmentation method, this gap may persist. Do the authors have any explanations or proposed solutions for this issue?
3. In Figures 6 and 7, all qualitative results appear to be zoomed-in versions. Could the authors provide full images, particularly for real-world natural images with dense haze? These images should represent natural haze rather than artificial fog.
4. The augmented dataset is based on NH-Haze20 and similar real-world datasets, which are limited and lack diversity (e.g., environmental and lighting conditions). How do the authors plan to address the reliance on these existing datasets?

---

> ### Author Response · Authors · 2024-11-23
> **Author Feedback 1**
>
> * **Questions about idealized assumptions of the physical scattering model for haze removal.**
>
> Our method does account for the inaccuracies of the idealized PSM model.  As detailed in Section 1 of the paper, the strong performance of PANet stems from its innovative two-step design.
> It employs a physically explainable (albeit simplified) model-based initial prediction, followed by a data-driven refiner that captures the residual errors from the idealized PSM model to mitigate the domain gap. The use of parametric augmentation in the ASM domain ensures the process remains both straightforward and physically interpretable. This approach  represents a novel integration of model-driven and data-driven strategies.
>
> * **Effectiveness of PANet on real-world natural hazy images and the discussion of using hazy images  generated by fog machine.**
>
> While hazes produced by fog machines may exhibit some differences from natural haze in real-world scenes, both types are governed by the same physical principles. Consequently, fog-machine-generated haze can effectively represent key characteristics of natural haze, such as its diffused, misty, and irregular nature, albeit with certain limitations compared to the complexity of real haze patterns. To address this gap, PANet employs a parametric augmentation scheme to diversify haze patterns, bridging the difference to some extent. As demonstrated in Tables 1 and 2, PANet enhances the performance of dehazing models not only on non-homogeneous haze datasets but also on the real-world natural haze dataset, RTTS. Additional qualitative results of dehazing models on the RTTS dataset are included in the supplementary materials.
>
> * **Qualitative results of full-size dehazed images.**
>
> We have included several full-size dehazed images of natural hazy scenes from the RTTS dataset in the supplementary materials. These examples demonstrate that PANet consistently delivers enhanced dehazing performance on real-world hazy images.
>
> * **Discussion of relying on existing datasets, the diversity of PANet, and the future plan.**
>
> This is an excellent point. While PANet significantly enhances the haze patterns in existing datasets of hazy-clean image pairs, thereby boosting the performance of current dehazing models, the diversity of the augmented dataset remains constrained by the environmental and lighting variations present in the original dataset, as noted by the reviewer. Addressing this limitation is crucial for further improving the comprehensiveness of augmented data.
>
> To this end, we are expanding the capabilities of the physics-guided augmentation strategy by employing a novel approach that involves "disentangling" and "transferring" haze patterns from target-domain hazy images. Importantly, this process does not require paired haze-free versions of the images. Instead, it maps the haze patterns to the scene content of haze-free images in a different domain within the physics-parametric framework. We believe this haze-transfer capability, combined with parametric augmentation, will significantly enhance the spectrum of haze augmentation, enabling more effective domain adaptation.

---

> > ### Comment · Reviewer_digo · 2024-11-26
> >
> > Dear Authors,
> >
> > Thank you for your detailed response. The rebuttal has addressed some of my concerns. However, I would like to point out a few remaining issues:
> >
> > 1. The data-driven refiner relies on paired images generated by a fog machine, while the parametric augmentation scheme is based on an idealized approximation. Both approaches introduce significant domain gaps that affect the method's generalizability.
> >
> > 2. The results presented in the supplementary material are not convincing, as the haze in the input images used for evaluation is sparse. This is in contrast to the NH-HAZE dataset, where the haze density is significantly higher, as shown in Figure 5. The domain gap between the synthetic and natural haze distributions appears to be the reason for the method's inability to handle dense haze effectively.
> >
> > Therefore, I tend to maintain my score.
> >
> > Best regards,

---

> > > ### Author Response · Authors · 2024-11-29
> > > **The domain gap between the NH-HAZE dataset and the natural haze**
> > >
> > > Although the fog-machine-generated haze in the NH-Haze dataset may differ in density from natural haze (e.g., often appearing thicker), our proposed PANet effectively addresses this by randomly varying the haze density (making it sparser or denser) and shifting its position to enhance haze diversity, as demonstrated in Figure 4.
> > >
> > > As previously noted, fog-machine-generated hazy images capture key characteristics of natural haze, such as diffusion, mistiness, and irregularity. By augmenting both sparse and dense haze patterns across diverse locations within the PSM parametric domain, PANet provides a practica; solution to bridge the gap between real-world natural haze, fog-machine-generated haze, and PSM-generated haze, ultimately improving dehazing performance.

---

> ### Author Response · Authors · 2024-11-25
>
> Dear digo,
>
> We have made every effort to address your concerns and sincerely hope our responses meet your expectations. We would greatly appreciate your feedback on our replies to help us further enhance the manuscript.

---

> ### Author Response · Authors · 2024-11-29
> **PANet's generalizability**
>
> (1) While the hazy images in NH-Haze, generated by a fog machine, may not entirely replicate real-world natural haze, they effectively capture essential characteristics such as diffusion, mistiness, and irregularity since the fog is governed by physics laws. This makes them significantly more representative of natural haze compared to hazy patterns generated solely by the physical scattering model (PSM). Moreover, the enriched haze patterns achieved through our parametric augmentation methods---such as variations in thickness, density, and spatial distribution---further reduce the disparity between fog-machine-generated haze and real haze.
>
> (2) Although our parametric augmentation is based on the PSM parametric domain, the subsequent Data-driven Haze Refinement (DHR) module bridges the gap between ASM-approximated haze and real haze, as detailed in our manuscript and responses.
>
> We have incorporated additional evaluation metrics, including BRISQUE, MUSIQ, and PAQ2PIQ, for the RTTS and Fattal's datasets, as presented in Table 2. However, since these quality assessment metrics are not specifically designed for image dehazing, they may not accurately reflect dehazing performance. Therefore, we further conducted a user study on the RTTS dataset, detailed in Appendix A.6, showing that PANet-enhanced results received 63  of the preference votes. Additionally, we provide extensive qualitative comparisons of dehazing models in real-world natural scenes, as shown in Figures 14 to 22. These experiments highlight PANet's effectiveness in handling hazy images in real-world natural scenes.

---

### Official Review · Reviewer_oeYm · 2024-11-03

**Soundness:** 3
**Presentation:** 3
**Contribution:** 3
**Rating:** 5
**Confidence:** 4

**Summary:**

Collecting real-world hazy-clean image pairs is particularly difficult and the authors tried to address this issue by proposing PANet. PANet can generate realistic hazy and clean training pairs, thus enhancing dehazing performance in real-world applications.

**Strengths:**

1. the key idea of performing parametric augmentation to generate additional haze patterns is good.
2. the experimental results are promising.
3. the paper is well-prepared and easy to follow.

**Weaknesses:**

1. the depth refinement module (DRM) is employed to refine the initial depth map, which means the depth estimation is not accurate enough in some cases. Have the authors attempted to utilize other methods of depth estimation which are more accurate?
2. the choice of baseline method lacks convincingness. The three baseline models, DW-GAN, Dehamer, and FocalNet are primarily utilized for synthetic data (i.e., SOTS-indoor, SOTS-outdoor). Can this method be applied to real-world dehazing models (e.g., RIDCP DAD)?
3. Comparisons with methods such as RIDCP DAD PTTD, which are oriented towards real image dehazing, are lacking. In addition, by observing the images, the qualitative results in Figure 7 contain some artifacts.
4. the parametric augmentation of haze is not flexible, can the value of $\alpha$ be continuous? What's the range of values for $\alpha$? What parameters were used in the experiments section?
5. the experiments section is not sufficiently comprehensive. For real-world hazy environments, only RTTS is tested and only NIQE and PIQE are adopted as the metrics.
6. some typos: e.g., Ln237.

**Questions:**

Please check the weaknesses part.

---

> ### Author Response · Authors · 2024-11-23
> **Author Feedback 1**
>
> **Table A. Cross-dataset comparisons of dehazing performance of haze augmentation methods, including RIDCP, D4, PTTD, and PANet.**
>
> | FocalNet | NH-Haze20 | NH-Haze21 | O-Haze | I-Haze | Average |
> |----------|-----------|-----------|--------|--------|---------|
> | Baseline | 20.31     | 16.51     | 18.28  | 15.29  | 17.60   |
> | +RIDCP   | 20.57     | 17.02     | 19.11  | **16.06** | 18.19   |
> | +D4      | 20.26     | 14.57     | 19.48  | 15.58  | 17.47   |
> | +PTTD    | 19.41     | 16.06     | 18.31  | 14.95  | 17.18   |
> | +PANet   | **20.76** | **17.87** | **20.64** | 15.41  | **18.67** |
>
> * **Effects of pre-trained depth estimation network and the proposed Depth-Refinement Network (DRM).**
>
> We address this comment from two perspectives:
>
> Depth Map Estimation: Our primary objective is not to estimate precise depth maps but rather to produce depth maps suitable for guiding realistic haze generation. Recognizing the inherent domain gap in any pre-trained depth estimator, we introduce a Depth Refinement Module (DRM) to mitigate this gap. Additionally, depth labels for the clean images in haze-clean image pairs are typically unavailable. To address this, we supervise the learning of the DRM using the discrepancy between the input hazy image and the regenerated hazy image, while keeping the ASM parameter maps unchanged in the "learning" mode.
>
> Fair Comparison: To ensure a fair comparison, we adopt the same depth estimator, RA-Depth [1], as used in RIDCP [2].
>
> * **The choice of baseline dehazed methods and comparison with existing haze augmentation methods.**
>
> In fact, DW-GAN, Dehamer, and FocalNet are not primarily trained on synthetic haze data. These three SOTA models were explicitly designed for both synthetic and real-world image dehazing tasks and have been evaluated on real-world, non-homogeneous haze datasets.
>
> Regarding the two methods you suggested, the authors of RIDCP [2] acknowledged its limitations in their paper, noting in the limitations section that “existing dehazing methods, including RIDCP, cannot process non-homogeneous (NH) haze well.” We attempted to re-train RIDCP using the NH-Haze20 training set; however, the model failed to converge and achieved a PSNR of only 12dB during training. Consequently, we chose not to include RIDCP as a baseline dehazing model in our experiments.
>
> Additionally, DAD [3] is a CycleGAN-based haze-transfer method, with its hazing model being a UNet-like architecture not specifically designed for non-homogeneous (NH) haze. Regarding rehazing augmentation capabilities, D4 [4] has been reported in [4] to outperform DAD. However, we have already demonstrated that PANet consistently outperforms D4.
>
> To further demonstrate the effectiveness of PANet, we perform cross-dataset validation comparisons, as shown in Table A. In this evaluation, dehazing models are trained on the baseline NH-Haze20 dataset and four expanded versions of NH-Haze20, augmented using PANet, RIDCP [2], D4 [4], and PTTD [5]. These models are then tested on four different datasets to assess their generalizability. The results indicate that the PANet-enhanced model consistently outperforms competing models across nearly all datasets, highlighting that PANet-augmented data effectively captures more diverse haze distributions.
>
>
> * **Discussion on artifacts in the qualitative results in Figure 7.**
>
> Due to the low resolution of these distant regions and the tendency for skies to be distorted by haze, fully recovering fine details remains challenging. Nevertheless, PANet significantly enhances dehazing performance by effectively eliminating haze artifacts compared to the baseline.
>
> * **Questions about the continuity, range, and parameters for the parametric augmentation in PANet.**
>
> In PANet, both atmospheric light map $A$ and haze density $\beta$ are treated as continuous parameters, where $A$ ranges in [0,1] and $\beta$ ranges in [0, 7.6] with an average value of 2.18. For haze-free regions, we inverse $A$ which is inverted to $1-A$ and $\beta$ is sampled from the range [0.6, 1.25] to generate haze patterns.
>
> * **Experiments on additional real-world hazy images.**
>
> We will include additional results and evaluation scores for real-world hazy environments in the revised manuscript and supplementary materials.
>
> * **Some typos: e.g., Ln237.**
>
> Thank you for your feedback, we have fixed the typos..
>
> [1] Ra-depth: Resolution adaptive self-supervised monocular depth estimation. In ECCV, 2022.
>
> [2] RIDCP: Revitalizing Real Image Dehazing via High-Quality Codebook Priors. In CVPR 2023.
>
> [3] Domain adaptation for image dehazing. In CVPR 2020.
>
> [4] Self-augmented unpaired image dehazing via density and depth decomposition. In CVPR, 2022.
>
> [5] Prompt-Based Test-Time Real Image Dehazing: A Novel Pipeline. In ECCV 2024.

---

> ### Author Response · Authors · 2024-11-25
>
> Dear oeYm,
>
> We have made every effort to address your concerns and sincerely hope our responses meet your expectations. We would greatly appreciate your feedback on our replies to help us further enhance the manuscript.

---

> > ### Comment · Reviewer_oeYm · 2024-11-25
> >
> > Dear Authors,
> >
> > Thank you for your response.
> > In fact, I am not entirely satisfied with your responses. Many of the experiments I suggested were not incorporated into the revised version. Additionally, I cannot fully agree with some of your explanations.
> >
> > Here are several main concerns from my view.
> >
> > 1. Authors claim that the goal of the DRM is to address the domain gap. After the training, the weights for DRM are fixed. How can DRM ensure its effectiveness across various $I_c$? Since PANet employs the physical scattering model to generate the $O_{ini}$, I still think the estimated $d(z)$ should be accurate. Otherwise, it will exert an influence on the estimation of $A$ and $\beta$. I also want to see the depth map before and after DRM.
> > 2. I checked the revised PDF and supplementary materials. However, I didn't find the additional results and other metrics I suggested. Maybe I missed them. Could you please directly point it out or highlight it?
> >
> > Best,
> >
> > Reviewer oeYm

---

> ### Comment · Reviewer_oeYm · 2024-11-26
>
> If the precise depth maps are not mandatory, why not directly utilize image $I_H$ to estimate the depth? The whole pipeline will be more flexible by adopting this strategy.
>
> Let me specify additional results and other metrics. Other real-world hazy datasets like Fattal's dataset from [1], and other no-reference image quality assessment metrics like MUSIQ, HyperIQA, paq2piq, etc.
>
> [1] Dehazing Using Color-Lines, TOG 2014

---

> ### Author Response · Authors · 2024-11-29
> **Discussion of DRM and visualization of the refined depth map and other feature maps**
>
> We employ the Data-driven Refinement Module (DRM) to address the domain gap between the pre-trained depth estimator and the haze-free images in the NH-Haze20 training set. Since our objective is to augment the existing hazy image dataset (NH-Haze20), we exclusively use the haze-free images provided within this dataset. The augmented dataset is then utilized to train dehazing models, which are subsequently evaluated across other datasets.
>
> Additionally, a refined depth map produced by DRM is illustrated in Figure 10 of the Appendix. Rather than aiming for precise depth estimation (as ground-truth depth maps are unavailable), DRM adjusts the depth map generated by the pre-trained depth estimator to make it "more suitable for real-haze generation." This refinement minimizes the discrepancy between the generated hazy image and its ground truth. As shown in Figure 2, the DRM learning process is supervised by the difference between the input hazy image and its regenerated version in the "learning mode" with the PSM parameter maps remaining unchanged.
>
> Our experiments demonstrate that this data-driven depth refinement effectively addresses the highly ill-posed problem of transmission map (or depth map) estimation, improving the fidelity of cyclic real-haze regeneration in the learning mode. In Table 4, we compare PANet against two variants: PANet without the depth estimator and DRM ("w/o depth and DRM") and PANet without DRM ("w/o DRM"). The results clearly indicate that PANet with DRM achieves the best performance.

---

> ### Author Response · Authors · 2024-11-29
> **Additional results and other metrics**
>
> We have incorporated additional evaluation metrics, including BRISQUE, MUSIQ, and PAQ2PIQ, for the RTTS and Fattal's datasets, as presented in Table 2. However, since these quality assessment metrics are not specifically designed for image dehazing, they may not accurately reflect dehazing performance. Therefore, we further conducted a user study on the RTTS dataset, detailed in Appendix A.6, showing that PANet-enhanced results received 63 $\%$ of the preference votes. Additionally, we provide extensive qualitative comparisons of dehazing models in real-world natural scenes, as shown in Figures 14 to 22. These experiments highlight PANet's effectiveness in handling hazy images in real-world natural scenes.

---

### Official Review · Reviewer_wXTj · 2024-11-04

**Soundness:** 3
**Presentation:** 3
**Contribution:** 2
**Rating:** 5
**Confidence:** 5

**Summary:**

This paper introduces the Physics-guided Parametric Augmentation Network (PANet), designed to improve real-world image dehazing.

PANet combines physics-based modeling with data-driven techniques to generate diverse hazy images, aiming to bridge the gap between synthetic and real-world hazy datasets.

By mapping haze characteristics into a parametric space, PANet can resample parameters and generate new, physically realistic hazy images.

**Strengths:**

- physics-guided + data-driven make sense.

- The physics-guided and parametric approach to generating realistic hazy images also makes sense.

**Weaknesses:**

- The progress of daytime dehazing or defogging has been significant over the past 10 years. These methods can handle many problems, particularly when the haze or fog is relatively thin. Non-uniform haze/fog is also not a significant issue, as many methods can handle it well. (If there is any disagreement, the paper should provide evidence of existing methods failing to deal with non-uniform haze.) The main challenge of dehazing arises when the haze/fog is significantly thick. Unfortunately, the proposed method does not address this thick haze/fog problem specifically, as evidenced by the results. Moreover, the proposed method has no specific mechanism or treatment in dealing with the thick haze/fog and its characteristics.

- The qualitative experimental results do not show that the proposed method outperforms the existing methods.
In Fig. 1 and 6, when the fog/haze is thick, the method still suffers from it and suffers from colour shift.

- The proposed method does not have any specific features that differentiate it from existing methods in terms of the haze/fog problem it aims to solve. The results presented in the paper could be achieved by existing methods, including non-deep learning methods, with comparable quality.

- Missing citation and dataset in [1]
[1] Structure Representation Network and Uncertainty Feedback Learning for Dense Non-Uniform Fog Removal

**Questions:**

1. For the colour shift issue, what is the reason?

2. It seems for the sky, white object and road, the results are not promising, what is the reason?

---

> ### Author Response · Authors · 2024-11-23
> **Author Feedback 1**
>
> **Table A. Cross-dataset comparisons of dehazing performance of haze augmentation methods, including RIDCP [1], D4 [2], PTTD [3], and PANet.**
>
> | FocalNet | NH-Haze20 | NH-Haze21 | O-Haze | I-Haze | Average |
> |----------|-----------|-----------|--------|--------|---------|
> | Baseline | 20.31     | 16.51     | 18.28  | 15.29  | 17.60   |
> | +RIDCP   | 20.57     | 17.02     | 19.11  | **16.06** | 18.19   |
> | +D4      | 20.26     | 14.57     | 19.48  | 15.58  | 17.47   |
> | +PTTD    | 19.41     | 16.06     | 18.31  | 14.95  | 17.18   |
> | +PANet   | **20.76** | **17.87** | **20.64** | 15.41  | **18.67** |
>
>
> * **Importance and challenges of non-uniform image dehazing and PANet’s approach to handling thick haze.**
>
> As highlighted in Table 1, existing SOTA dehazing methods struggle to handle non-uniform haze, resulting in subpar performance. The authors of RIDCP [1] also acknowledged this limitation in their paper (see its "limitations" section), stating that “existing dehazing methods, including RIDCP, cannot process non-homogeneous haze well.” This underscores the ongoing challenge of effectively addressing non-uniform image dehazing.
>
> In Figure 6, we showcase dehazed images under thick haze conditions. The "Baseline" model (i.e., a SOTA model trained on the baseline dataset) fails to perform adequately in these scenarios, whereas PANet-enhanced models demonstrate significant improvements. PANet effectively augments hazy images containing thick haze, as illustrated in Figures 5(b) and 5(d). These augmented images expand the training datasets with more diverse haze conditions, leading to superior dehazing performance in challenging environments.
>
> To further demonstrate the effectiveness of PANet, we perform cross-dataset validation comparisons, as shown in Table A. In this evaluation, dehazing models are trained on the baseline NH-Haze20 dataset and four expanded versions of NH-Haze20, augmented using PANet, RIDCP [1], D4 [2], and PTTD [3]. These models are then tested on four different datasets to assess their generalizability. The results indicate that the PANet-enhanced model consistently outperforms competing models across nearly all datasets, highlighting that PANet-augmented data effectively captures more diverse haze distributions.
>
> * **Qualitative comparisons between PANet and existing haze augmentation methods.**
>
> Thanks for the comment. Besides the qualitative comparisons between the PANet-enhanced model and the baseline, We have provided additional quantitative comparisons with exiting haze augmentation in Table A. We also provide the qualitative comparisons between PANet and existing haze augmentation methods in the section Appendix A.2.
>
> * **Difference between PANet and existing methods and additional comparison with a non-deep learning method**
>
> In our paper, we compared several representative haze augmentation methods, including PSM-based synthesis, PSM with adverse lighting conditions (RIDCP [1]), and the CycleGAN-based D4 [2], demonstrating that PANet significantly outperforms these approaches. We appreciate the reviewer for highlighting the recent ECCV 2024 non-deep learning method, PTTD [3], which generates visual prompts to enhance dehazing performance. However, as PTTD was introduced the same week as our ICLR submission, we were unable to include a comparison in the initial paper.
>
> That said, as shown in Figure 8 in the updated paper, the visual prompts generated by PTTD often exhibit patch-wise artifacts that deviate significantly from real-world haze distributions. Consequently, applying the augmented haze patterns produced by PTTD to train dehazing models yields poor performance in real haze scenarios, achieving only 8–10 dB (see Table 1 in [3]). Therefore, we respectfully disagree with the claim that existing haze augmentation methods achieve performance comparable to PANet.
>
> As detailed in the paper, the strong performance of PANet stems from its innovative two-step design. First, it employs a physically explainable (albeit simplified) model-based initial prediction, followed by a data-driven refiner that captures the residual errors from the simplified PSM model. The use of parametric augmentation in the ASM domain ensures the process remains both straightforward and physically interpretable. This approach is not merely a combination of existing methods but represents a novel integration of model-driven and data-driven strategies.
>
> [1] RIDCP: Revitalizing Real Image Dehazing via High-Quality Codebook Priors. In CVPR 2023.
>
> [2] Self-augmented unpaired image dehazing via density and depth decomposition. In CVPR, 2022.
>
> [3] Prompt-Based Test-Time Real Image Dehazing: A Novel Pipeline. In ECCV 2024.

---

> ### Author Response · Authors · 2024-11-23
> **Author Feedback 2**
>
> * **Discussion on the paper: Structure Representation Network and Uncertainty Feedback Learning for Dense Non-Uniform Fog Removal [4]**
>
> Thank you for bringing this interesting work to our attention. We have cited this paper and explore how this real-world dataset could contribute to improving dehazing or potentially support future extensions beyond dehazing in the updated paper.
>
> * **Issue of colour shift in Figure 1 and 6.**
>
> Dehazing images with thick haze is a challenging inverse problem, as scene content is often heavily occluded by haze, and certain image regions may appear overexposed. Due to insufficient training data, existing state-of-the-art (SOTA) dehazing models frequently produce results with unsatisfactory visual quality and noticeable artifacts. While PANet-enhanced results may still exhibit slight color shifts compared to the ground truth, they are significantly cleaner and more visually appealing than those produced by SOTA baselines, demonstrating a substantial improvement.
>
> * **Dehazed results for the sky, white object, and road.**
>
> These regions are often overexposed and exhibit low contrast, providing insufficient visual cues for dehazing models to achieve effective restoration. The problem is exacerbated by limited training data, as the scarcity of diverse examples restricts the ability of dehazing models to learn effectively. PANet addresses this issue to some extent by generating diverse haze patterns, ranging from thick to thin haze, over these overexposed regions, as evidenced by the results.
>
> [4] Structure Representation Network and Uncertainty Feedback Learning for Dense Non-Uniform Fog Removal. In ACCV 2022.

---

> ### Author Response · Authors · 2024-11-25
>
> Dear WXTj,
>
> We have made every effort to address your concerns and sincerely hope our responses meet your expectations. We would greatly appreciate your feedback on our replies to help us further enhance the manuscript.

---

### Official Review · Reviewer_ghxu · 2024-11-12

**Soundness:** 2
**Presentation:** 3
**Contribution:** 2
**Rating:** 5
**Confidence:** 4

**Summary:**

The paper addresses the real data-scarcity issue for dehazing: that real-world haze is often dense and non-homogeneous, which is difficult to synthesize using traditional image formation models. The proposed haze data augmentation technique (PANet) adopts a hybrid approach, combining the strengths of both data-driven or physics-based methods. It first estimates haze parameters from clean-hazy image pairs using Haze to Parameter Mapper. In the Parameter to Haze mapper: it leverages physics-guided scattering model to generate initial hazy images. It further incorporates a Data-driven Haze Refiner (DHR) to refine this initial hazy images to enable better realism and accuracy.

**Strengths:**

The paper addresses a practical problem in dehazing: real-world haze is often dense and non-homogeneous, which is difficult to synthesize purely using physical scattering image formation models.

The HPM+PHM cyclic approach for unsupervised learning of intermediate haze parameters is practically effective.

Applying the proposed augmentation on selected Dehazing methods leads to notable improvement in dehazing quality on real images and few synthetic test images.

The approach is data efficient, in which it can be trained on a small dataset of as few as 50 images. The hybrid formulation leads to fewer unwanted artifacts than GAN based augmentation approaches.

**Weaknesses:**

Limited technical novelty: The approach is derived from existing, established methods for cyclic image-to-image mapping, specifically built upon CycleGAN.

Dataset limitations: The analysis and evidence for validating the idea are limited, as the validation relies on a small real-world dataset (NH-Haze20) for training, with only 50 training pairs and 5 testing pairs. This limited dataset size may restrict the generalizability and effectiveness of PANet in handling the diversity of real-world haze conditions.

Computational footprint and scalability: PANet is a relatively complex architecture with multiple components, including encoders, decoders, a depth refinement module, and a data-driven haze refiner. This complexity requires significant FLOPs and increases the computational cost and training time compared to simpler augmentation techniques. Additionally, how well PANet scales to larger datasets (on the order of 10^4 to 10^6 images) should be discussed.

Few writing quality issues: There are some quality issues in writing, such as an equation reference error on line 238 and typos like “pixel-wisely” on line 307.

Outdoor vs. indoor image improvement: The improvement on outdoor hazy images appears to be higher than on indoor hazy images. This observation should be discussed further.

Qualitative results clarity: It is not clear which of the three dehazing models was used to generate qualitative results, such as those in Figs. 6 and 7.

Choice of augmentations: Some choices of augmentation, such as “reverse its haze location,” seem less realistic, as they are opposite to the general nature of haze (which typically increases with distance). It would be interesting to analyze the effect of excluding such augmentations.

Dependency on DHR: The results in Table 3 suggest that the entire approach fails if the Depth Haze Refiner (DHR) is not included, which is surprising and questions the method’s utility. There should be an analysis with quantitative and qualitative results on the effect of the Depth Estimator and DRM on the performance. Additionally, extensive visualizations showing the outputs of the depth estimator, DRM, beta(z), and final t(z) are recommended.

Reliance on pre-trained depth estimator: PANet relies on a pre-trained depth estimator (RA-Depth) to estimate depth maps from clean images, which may pose a potential weakness. This estimator may not generalize well to unseen images, especially those with characteristics different from its training data. This generalization issue may not always be addressable by training a DRM, which could lead to inaccurate depth estimations and negatively impact the accuracy of the physical scattering model used in PANet, affecting the realism of the generated hazy images.

Baseline model performance: The results of three baseline dehazing models on real images from the RTTS dataset appear to be quite poor, with significant artifacts. It would be interesting to know whether any existing dehazing model can yield reasonable results on the RTTS dataset.

Selection of dehazing models: How were the three dehazing models selected? Additionally, it might be interesting to analyze any improvements observed when using other recent dehazing models.

Risk of overfitting: The potential for overfitting needs to be carefully considered. While PANet shows improvements in dehazing performance on a few similar datasets and one additional real dataset, the use of augmented data can increase the risk of overfitting, especially with a limited original dataset.

Additional metrics: Including additional no-reference metrics, such as FADE, BRISQUE, NIMA, and US, for the RTTS datasets would enable a fuller comparison with RIDCP (Wu et al., 2023).

Evaluation on popular benchmarks: The proposed approach could also be evaluated on popular dehazing benchmarks like the SOTS-Outdoor and SOTS-Indoor datasets.

Evaluation under challenging conditions: Optionally, it might be interesting to test PANet under extremely challenging haze conditions, such as dense fog or heavy smog.

**Questions:**

Please address weaknesses above.

---

> ### Author Response · Authors · 2024-11-23
> **Author Feedback 1**
>
> * **Difference between PANet and CycleGAN.**
>
> As mentioned in section 1, PANet dose not rely on CycleGAN, which is known for its instability and inability to produce controllable results. In contrast, PANet is a robust haze augementation network that enables pixel-wise adjustments of haze conditions to augment realistic hazy images not provided in the training set.
>
> * **Discussion of dataset limitations.**
>
> To ensure a fair comparison, we use the NH-Haze20 training set to optimize both PANet and the dehazing models. To verify the generalizability of PANet, we perform cross-dataset evaluations by testing dehazing performance on the NH-Haze21, O-Haze, and I-Haze datasets. Additionally, we evaluate dehazing performance on the RTTS dataset, which contains 4,322 hazy images collected from real-world scenarios without corresponding haze-free images. These experiments demonstrate the generalizability and effectiveness of PANet.
>
> * **Discussion of computational footprint and plan to scale to larger datasets**
>
> We provide computational footprint of PANet in the section Appendix. PANet consists of 3M parameters and requires 23 GFLOPs, with an inference time of 25 ms for 256×256 images. In PANet, we focus on increasing haze diversity by utilizing haze-free images from the NH-Haze20 training set. Since PANet can be successfully optimized with limited training data, it has the potential to be applied to larger datasets for further augmentation. However, as existing real-world image dehazing datasets only provide limited training samples, we are unable to conduct experiments using larger datasets.
>
> * **Writing issues in line 238.**
>
> Thank you for your feedback, we have revised it.
>
> * **Dehazing performance for outdoor and indoor hazy images.**
>
> In PANet, we augment training pairs using outdoor haze-free images from the NH-Haze20 dataset, resulting in greater performance improvements on outdoor hazy images compared to indoor hazy images.
>
> * **Dehazing models in Figures 6 and 7.**
>
> We adopt dehazing models, including DE-GAN, DeHamer, and FocalNet, as listed on the left side of Figures 6 and 7.
>
> * **Discussion of using pre-trained depth estimator.**
>
> We address this comment from two perspectives:
>
> Depth Map Estimation: Our primary objective is not to estimate precise depth maps but rather to produce depth maps suitable for guiding realistic haze generation. Recognizing the inherent domain gap in any pre-trained depth estimator, we introduce a Depth Refinement Module (DRM) to mitigate this gap. Additionally, depth labels for the clean images in haze-clean image pairs are typically unavailable. To address this, we supervise the learning of the DRM using the discrepancy between the input hazy image and the regenerated hazy image, while keeping the ASM parameter maps unchanged in the "learning" mode.
>
> Fair Comparison: To ensure a fair comparison, we adopt the same depth estimator, RA-Depth [1], as used in RIDCP [2].
>
> * **The choice of baseline dehazed methods and their performance on the RTTS dataset.**
>
> We adopt state-of-the-art image dehazing models for experiments. Since RTTS consists of hazy images collected from natural scenes without corresponding haze-free images, existing dehazing methods struggle to fully recover high-quality clean images. However, PANet significantly improves baseline dehazing performance, producing visually more satisfactory results.
>
> * **Discussion of overfitting.**
>
> PANet can generate diverse hazy images to augment existing datasets, helping to prevent dehazing models from overfitting. As shown in Tables 1 and 2, dehazing models without PANet tend to overfit on the training sets, leading to unsatisfactory performance on the NH-Haze21, O-Haze, I-Haze, and RTTS datasets. In contrast, PANet-enhanced dehazing models consistently improve upon baseline models, delivering better dehazing results.
>
> * **Evaluation on the SOTS-Outdoor and SOTS-Indoor datasets.**
>
> Since PANet focuses on improving dehazing performance on real-world dehazing datasets, we do not use the SOTS-Outdoor and SOTS-Indoor datasets, as they consist of homogeneous synthetic haze.
>
> * **Evaluation under challenging conditions.**
>
> NH-Haze20 and NH-Haze21 datasets contain hazy images with thick haze. PANet consistently improve dehazing performance on these two datasets. We have demonstrate qualitative comparisons under thick haze in Figure 6.
>
> [1] Ra-depth: Resolution adaptive self-supervised monocular depth estimation. In ECCV, 2022.
>
> [2] RIDCP: Revitalizing Real Image Dehazing via High-Quality Codebook Priors. In CVPR 2023.

---

> > ### Author Response · Authors · 2024-11-25
> >
> > Dear ghxu,
> >
> > We have made every effort to address your concerns and sincerely hope our responses meet your expectations. We would greatly appreciate your feedback on our replies to help us further enhance the manuscript.

---

> ### Author Response · Authors · 2024-11-27
> **Dehazing performance of PANet without reverse haze augmentation on NH-Haze20 test set.**
>
> | FocalNet | Baseline | PANet (w/o reverse haze) | PANet
> |----------|-----------|-----------|--------
> | PSNR| 20.31     | 20.20   | **20.76**
>
> PANet without reverse haze augmentation reduces the haze diversity in the augmented images, resulting in worse performance compared to the final version of PANet.

---

> ### Author Response · Authors · 2024-11-27
> **Additional evaluation metrics for RTTS and Fattal's datasets.**
>
> We have included additional evaluation metrics, including BRISQUE, MUSIQ, and PAQ2PIQ, for the RTTS and Fattal's datasets in Table 2. To further demonstrate PANet's capability in improving dehazing performance in real-world natural scenes, we conducted a user study on the RTTS dataset, detailed in Appendix A.6, showing that PANet-enhanced results received 63% of the preference votes. Additionally, we provide extensive qualitative comparisons of dehazing models in real-world natural scenes, as shown in Figures 14 to 22. These experiments highlight PANet's effectiveness in handling hazy images in real-world natural scenes.

---

### Note · Authors · 2025-01-22

I have read and agree with the venue's withdrawal policy on behalf of myself and my co-authors.